# Histidine modulates amyloid-like assembly of peptide nanomaterials and confers enzyme-like activity

Ye Yuan[1,2,3], Lei Chen[2], Lingfei Kong[4], Lingling Qiu[5], Zhendong Fu[1], Minmin Sun[2], Yuan Liu[3], Miaomiao Cheng[3], Saiyu Ma[3], Xiaonan Wang[2], Changhui Zhao[1], Jing Jiang[2], Xinzheng Zhang [4], Liping Wang[1] ✉ & Lizeng Gao [2,3] ✉

Amyloid-like assembly is not only associated with pathological events, but also leads to the development of novel nanomaterials with unique properties. Herein, using Fmoc diphenylalanine peptide (Fmoc–F–F) as a minimalistic model, we found that histidine can modulate the assembly behavior of Fmoc–F–F and induce enzyme-like catalysis. Specifically, the presence of histidine rearranges the β structure of Fmoc–F–F to assemble nanofilaments, resulting in the formation of active site to mimic peroxidase-like activity that catalyzes ROS generation. A similar catalytic property is also observed in Aβ assembled filaments, which is correlated with the spatial proximity between intermolecular histidine and F-F. Notably, the assembled Aβ filaments are able to induce cellular ROS elevation and damage neuron cells, providing an insight into the pathological relationship between Aβ aggregation and Alzheimer's disease. These findings highlight the potential of histidine as a modulator in amyloid-like assembly of peptide nanomaterials exerting enzyme-like catalysis.

The ability of proteins and peptides to assemble into amyloid fibrils has been recognized as having a strong association with severe neurodegenerative diseases[1,2]. The amyloid-like assembly shares a common cross-β structure, in which the β-strand segments align perpendicular to the long fibril[3,4]. Such type of assembly is driven by backbone hydrogen bonding and side-chain interactions (e.g., π–π stacking, hydrophobic interaction, and van der Waals)[5]. The Aβ is one of the typical proteins that can assemble into amyloid aggregates with a cross β pattern and frequently appear in many neurodegenerative diseases such as AD and Parkinson's disease (PD)[6–8]. Studies on Aβ assembly facilitate our understanding of the pathogenesis of AD, possibly providing potential targets for diagnosis and intervention[9]. Aβ1–42, which is more hydrophobic and more prone to aggregation, is recognized as a more important contributor to AD development than Aβ1–40, which shows less toxic effect. However, the fundamental role and implication of amyloid assembly of Aβ fragments have not yet been fully elucidated in AD pathogenesis.

The study on amyloid-like assembly in living systems has also promoted the exploration of engineering short peptide assemblies as emerging nanomaterials with exceptional properties[10,11]. Many small amyloid-forming peptides have been identified from parent proteins/polypeptides. For example, the diphenylalanine (F–F) as the shortest peptide able to self-assemble is identified by dissecting the structural information of Aβ1–42 polypeptide[12,13]. Due to its chemical simplicity and excellent self-assembly capacity, the F–F peptide has been employed in a wide range of nanostructures, such as nanotubes,

[1]Key Laboratory for Molecular Enzymology and Engineering, School of Life Sciences, Jilin University, Changchun 130012, China. [2]CAS Engineering Laboratory for Nanozyme, Key Laboratory of Biomacromolecules, Institute of Biophysics, Chinese Academy of Sciences, Beijing 100101, China. [3]Nanozyme Medical Center, School of Basic Medical Sciences, Zhengzhou University, Zhengzhou 450001, China. [4]National Laboratory of Biomacromolecules, CAS Center for Excellence in Biomacromolecules, Institute of Biophysics, Chinese Academy of Sciences, Beijing 100101, China. [5]Key Laboratory of Animal Genetics and Breeding and Molecular Design of Jiangsu Province, Yangzhou University, Yangzhou, China. ✉e-mail: wanglp@jlu.edu.cn; gaolizeng@ibp.ac.cn

nanowires, nanoarrays, nanospheres, and microtubes[14]. Given their unique mechanical and physical properties, these materials have promising application potential, including as templates, piezoelectrical and optical biosensors, and nanomedicines[15–20]. In addition, modification of chemicals or addition of co-assembly molecules regulates F–F assembly. For example, modification at the amino terminus of F–F with hydrophobic fluorenylmethoxycarbonyl (Fmoc)[21] facilitates the formation of β-sheet-based fibrous-hydrogel[22]. The assembly has been mostly conducted in organic solvents, such as dimethyl sulfoxide (DMSO), or in a mixed system of water and strong polar solvents, such as hexafluoroisopropanol (HFIP). In contrast, the strategy is currently limited to modulating the assembly behavior of these short peptides in aqueous conditions, which enables the assembly to more closely resemble those occurring in physiological environments.

In our study, we report that amino acids have the potential to modulate the assembly behavior of Fmoc–F–F under aqueous conditions through a sonication-standing process. Specifically, His was found to promote the aggregation of Fmoc–F–F dipeptides from nanorods into nanofilaments through electrostatic interactions and hydrogen bonding, leading to the β structure rearrangement in the Fmoc–F–F amyloid-like assembly. Furthermore, the resulting Fmoc–F–F (His) filaments exhibited peroxidase (POD)-like activity that catalyzes ROS generation, with the imidazole group of His serving as the active site. Building upon the understanding of the interaction mode between His and F–F dipeptide, we found that Aβ1–42 peptide-assembled nanofilaments exhibited POD-like activity which increased as the aggregation time prolonged. Importantly, Alphafold2-assisted structure analysis demonstrated that the intermolecular interaction between His and F–F contributed to the formation of active sites, which may enable Aβ1–42 filaments as a nanozyme to work in a physiological environment. Cellular and in vivo experiments demonstrated that Aβ1–42 filaments induced ROS toxicity to damage neuron cells, which provides an insight into the causal link between Aβ aggregation and AD pathogenesis.

## Results

### His regulates Fmoc–F–F (His) assembly from nanorods to nanofilaments

To understand the general effects of amino acids on Fmoc–F–F assembly, natural amino acids were mixed with Fmoc–F–F under aqueous condition (Fig. 1a). When amino acids absent, the Fmoc–F–F dipeptides self-assembled into stacked nanorods in aqueous solution and existed as milky white turbid liquid without DMSO or HFIP (Fig. 1b, c, f). The short thick nanorods shown in Fig. 1b (scanning electron microscopy, SEM) and Fig. 1c (transmission electron microscopy, TEM) were 3–6 μm long and 200–400 nm wide. In contrast, a clear hydrogel consisting of nanofilaments was formed with the addition of His to the Fmoc–F–F co-assembly (Fig. 1d–f). Nanorods were converted into slender filaments (-20 nm wide) (Fig. 1d SEM, Fig. 1e TEM). The high-angle annular dark field (HAADF)-TEM further confirmed the morphology of nanofilaments (Supplementary Fig. 1a). Negative-staining TEM showed that a single filament was composed of many finer bundles (Supplementary Fig. 1b). The smooth surface of the filaments (Rq of 6.05 nm) was observed via atomic force microscopy (AFM) (Supplementary Fig. 1c, d). Rheological measurements of dynamic strain sweeps indicated the occurrence of a phase change from turbid liquid to hydrogel (G″ > G′ → G′ > G″), which was invariant with increasing frequency, demonstrating typical hydrogel properties (Fig. 1g). Encouraged by this phenomenon, we systematically studied the co-assembly of Fmoc–F–F with amino acids.

In addition to His, Asparagine (Asn) (Supplementary Fig. 2a), Asparticacid (Asp) (Supplementary Fig. 2b), Serine (Ser) (Supplementary Fig. 2c), Lysine (Lys) (Supplementary Fig. 2d), Arginine (Arg) (Supplementary Fig. 2e), Proline (Pro) (Supplementary Fig. 2f), and Leucine (Leu) (Supplementary Fig. 2g) could regulate the assembly of

Fmoc–F–F into filaments. However, the filaments were not as extensive as those produced by His, and some nanorods were still present in the co-assembly systems. Furthermore, the addition of Pro appeared to change the structure and shape (block-shape) of the Fmoc–F–F nanorods. In contrast, Threonine (Thr), Tyrosine (Tyr), and Cysteine (Cys) could not regulate Fmoc–F–F into filaments, although the nanorods showed orderly aggregation into spheres, as observed in the TEM images of Fmoc–F–F (Tyr) (Supplementary Fig. 2h), Fmoc–F–F (Thr) (Supplementary Fig. 2i), and Fmoc–F–F (Cys) (Supplementary Fig. 2j). Fmoc–F–F (Cys) showed the greatest degree of sphere formation. The morphologies of the other Fmoc–F–F-amino acid co-assemblies were very similar to Fmoc–F–F (Supplementary Fig. 2k–s). Taken together, in the Fmoc–F–F-amino acid co-assembly systems, positively charged amino acids regulated nanorods into filaments due to electrostatic interactions, whereas the polar uncharged amino acids retained the nanorods to varying degrees. The co-assembly conditions are summarized in Supplementary Table 1. As Fmoc–F–F (His) showed the greatest degree of fibrillogenesis, it was used for the following research.

To obtain the optimal co-assembly conditions for Fmoc–F–F (His), we investigated sonication time, dipeptide concentration, amino acid concentration, and dipeptide to amino acid molar ratio. Only a few filaments were produced from aggregated nanorods with a short sonication time (10 min). The degree of depolymerization was intensified under longer sonication time (20 min), but too long sonication time (45 min) resulted in filament entanglement (Supplementary Fig. 3). Thus, the Fmoc–F–F (His) filaments showed the highest degree of expansion and uniform dispersion after sonication treatment for 30 min. When the concentration of Fmoc–F–F was fixed to 2 mg mL$^{-1}$, low concentrations of His (2 mg mL$^{-1}$) had little impact on the Fmoc–F–F structure, with only a few filaments peeling from nanorods. At high concentrations (30 mg mL$^{-1}$), His intensified the entanglement of filaments. The His concentration in the range of 18–20 mg mL$^{-1}$ was suitable for the formation of filaments (Supplementary Fig. 4). When the concentration of His was fixed to 20 mg mL$^{-1}$, too low (0.5 mg mL$^{-1}$) or too high (>3 mg mL$^{-1}$) concentrations of Fmoc–F–F caused the formation of denser filaments (Supplementary Fig. 5). The resistance value of water was also considered (Supplementary Fig. 6).

To understand the chemical group by which His affects the Fmoc–F–F structure, different side-chain modifications of His were used for co-assembly with dipeptides. Fmoc–F–F (Boc-His (Trt)-OH) (Supplementary Fig. 7a) and Fmoc–F–F (N-Acetyl-L-His) (Supplementary Fig. 7b) showed similar structures to Fmoc–F–F. The difference was that Fmoc–F–F (Boc-His (Trt)-OH) formed thicker nanorods than Fmoc–F–F (N-Acetyl-L-His), which may be related to the asymmetric N-ring structure (like Fmoc–F–F (Pro)). Acetyl-modified His was unable to form filaments with Fmoc–F–F due to missing amino groups. Nanorod leaves were obtained from Fmoc–F–F (Fmoc-His) (Supplementary Fig. 7c, d). These results demonstrated that both imidazole and amino groups played important roles in the formation of filaments. Moreover, Fmoc–F–F (His-OMe) (Supplementary Fig. 6e, f) showed reduced formation of filaments, indicating that the carboxyl group promoted the dissociation of nanorods and formation of filaments.

### Cryo-EM characterization reveals Fmoc–F–F (His) filaments composed of spindle structure

To further understand the structure of Fmoc–F–F (His) formed nanofilaments, Cryo-EM was conducted to characterize local features (Fig. 2). As shown in Fig. 2a, the diameter of Fmoc–F–F nanofilaments was around 20 nm. However, when the image was amplified, the filament showed heterogeneous diameters with a shape-like spindle structure which is narrow at two ends (yellow and blue circles) and wide in the middle part (red circle). Furthermore, two-dimensional structure analysis performed and demonstrated that the two ends

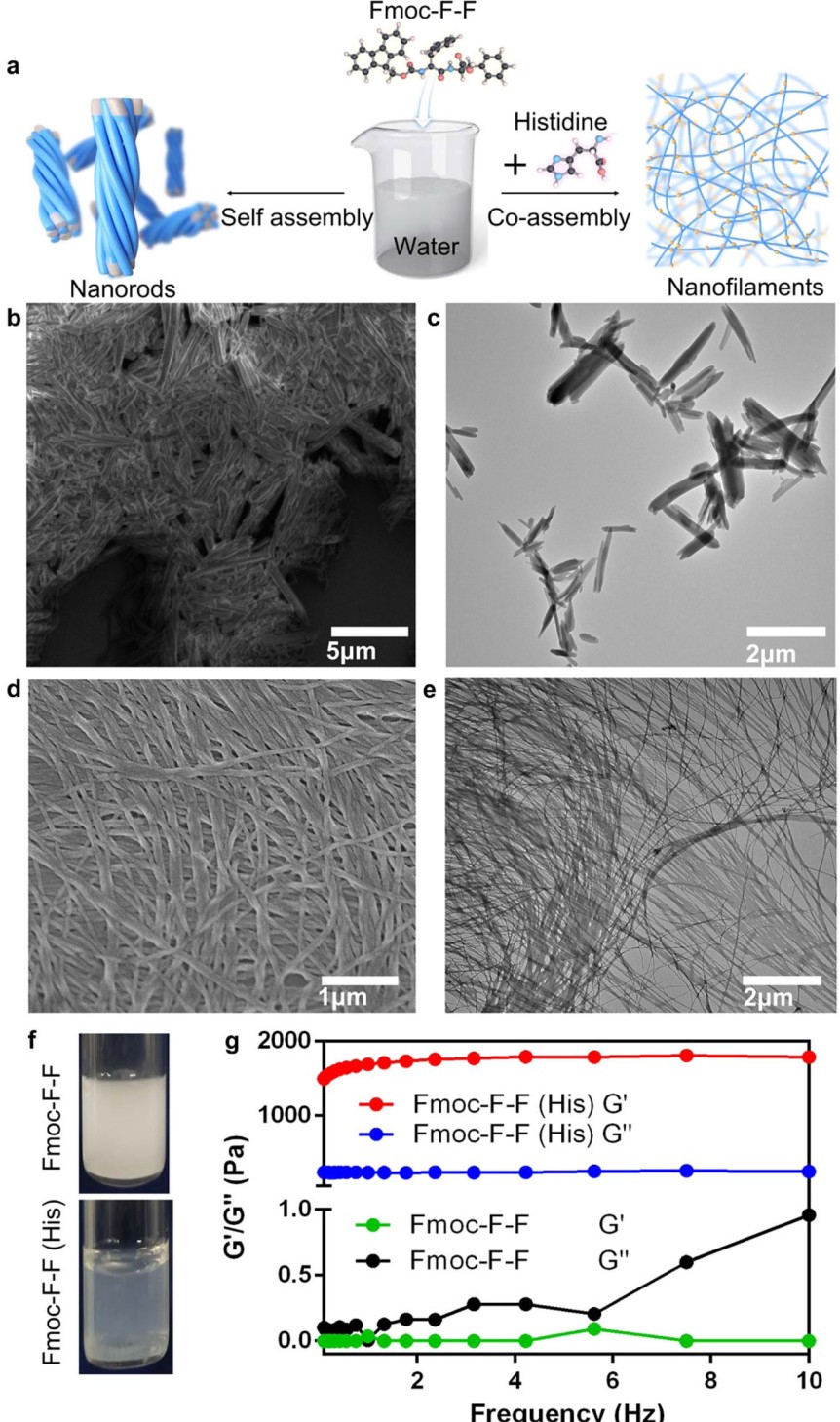

**Fig. 1 | Phase change of Fmoc–F–F in His aqueous solution. a** Schematic of phase change of co-assembly Fmoc–F–F by His. **b**, **c** SEM and TEM of self-assembled Fmoc–F–F in pure water. Stubby nanorods were obtained. **d**, **e** SEM and TEM of co-assembled Fmoc–F–F (His). Thin and long filaments transformed from nanorods were obtained due to the presence of His. **f** milky white turbid liquid formed by Fmoc–F–F and clear hydrogel formed by Fmoc–F–F (His). **g** Occurrence of phase transition based on rheological measurements of dynamic frequency sweeps of Fmoc–F–F (His). Three times each experiment was repeated independently with similar results. Representative images are shown. Source data are provided as a Source Data file.

were at 13.2 nm and the middle part was around 18.5 nm in the spindle unit (with the length -100 nm) in the corresponding circles in Fig. 2a (Fig. 2b). In the red circle of Fig. 2b, 20 bundles which with the diameter around 0.58–0.75 nm were displayed, indicating that the nanofilaments were parallel aligned by tiny bundles in the middle part. In contrast, in the yellow and blue circles of Fig. 2b, the entangled bundles were observed. In particular, thick bundles seemed to be bifurcated into thin bundles, resulting in the bundle winding pitch at about 132 Å. These features indicated that the two ends of the spindle unit might be transitional junctions to connect short spindles into a long nanofilament. The spindle structure of Fmoc–F–F (His) nanofilaments was confirmed by the negatively stained TEM images (Fig. 2c).

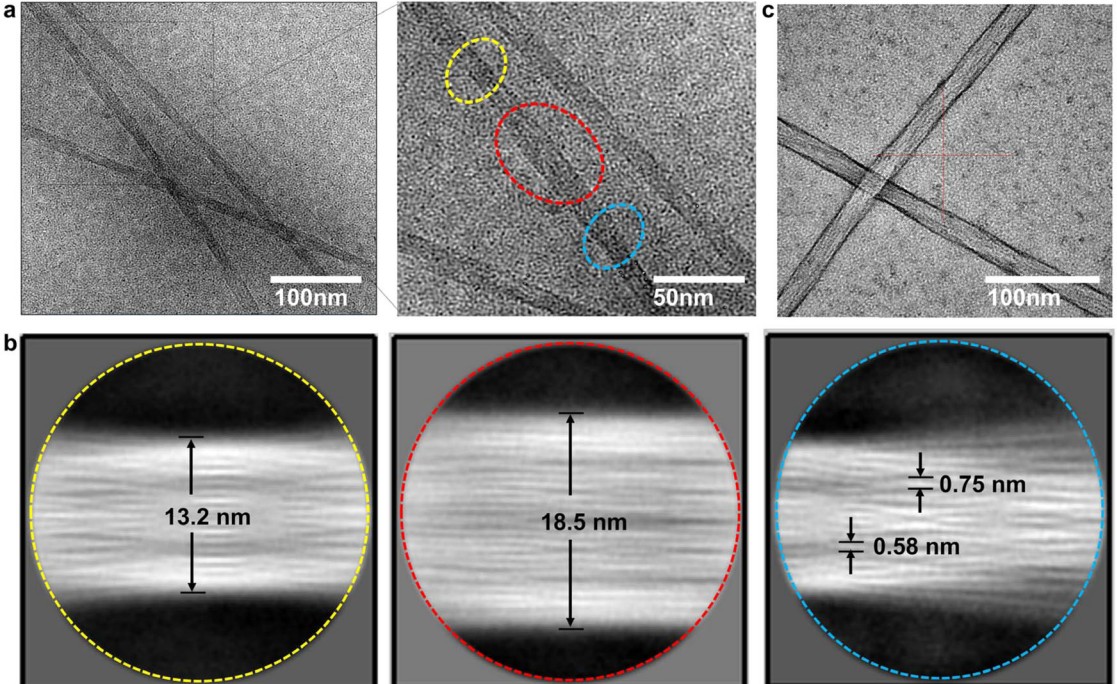

**Fig. 2 | Cryo-EM characterizations for Fmoc–F–F (His). a** Cryo-EM image of Fmoc–F–F (His) showing primitive spindle filament with heterogeneous diameter. **b** Two-dimensional classification analysis of Fmoc–F–F (His) showing parallel alignment and entangled bundles in spindle structure. The images correspond to three morphologies (dashed yellow circle, dashed red circle, and dashed blue circle) of boundles in **a**, respectively. **c** Negatively stained Cryo-EM image of Fmoc–F–F (His) using uranium acetate confirming the spindle structure of nanofilaments. Three times each experiment was repeated independently with similar results. Representative images are shown.

Based on the above characterizations; we speculated that the whole nanofilament is assembled by forming connective spindle units, which consist of the main structure with parallel bundles and entangled bundles at the ends. Overall, Cryo-EM further confirmed that His changed the aggregation of Fmoc–F–F and formed co-existing tangled nanofilaments from stacked nanorods.

### His remodels intermolecular interactions for Fmoc–F–F assembly into nanofilament

To understand the interactions between Fmoc–F–F and His, spectral characterizations were conducted. First, based on ultraviolet-visible spectroscopy (UV–VIS) (Supplementary Fig. 8), Fmoc–F–F (His) showed strong absorption peaks at 260 nm and 300 nm, corresponding to the absorption of benzene ring and Fmoc, respectively. The formation of these peaks in aqueous solution was due to the combination of Fmoc–F–F and His. Consistently, Fourier-transform infrared spectroscopy (FTIR) (Supplementary Figs. 9 and 10) identified that the benzene band at 1450 cm$^{-1}$, representing the C=C vibration of the benzene ring, significantly decreased in the presence of His. In addition, Fmoc–F–F exhibited stronger C–N stretching (1030 cm$^{-1}$) and N-H (739 cm$^{-1}$, 3300 cm$^{-1}$) bending compared with Fmoc–F–F (His) and His[23]. Weakening of the Fmoc–F–F (His) amines indicated the formation of non-covalent interactions. The red shift of C=O (1630 cm$^{-1}$) implicated the involvement of –COOH in hydrogen bonding. Furthermore, the redshift of –OH (3400 cm$^{-1}$) under different amounts of Fmoc–F–F (His), indicated the formation of intermolecular hydrogen bonds. Third, X-ray diffraction (XRD) of Fmoc–F–F (His) was performed to investigate the interaction between His and Fmoc–F–F. The crystal peak diffraction intensity was high, sharp, and scattered, indicating a good crystal state. In addition, the fewer peaks of Fmoc–F–F (His) accompanied by a shift in the diffraction peaks suggested that His may enter the original crystal lattice and cause distortion. Strong diffraction peaks at 21° and 24° were formed

(Supplementary Fig. 11). Fourth, the chemical environment of Fmoc was tracked by fluorescence spectroscopy. Neither F–F (H$_2$O) nor F–F (His) showed emission absorption (Supplementary Fig. 12). However, under 280-nm excitation, Fmoc–F–F (His) showed a strong emission peak at ~312 nm, with a slight blue shift compared to Fmoc–F–F (320 nm) (Supplementary Fig. 13). This may be due to His weakening the interaction of π–π stacked Fmoc but enhancing the local effect of dipeptide. To show its fluorescence characteristics more intuitively, three-dimensional (3D) fluorescence analysis was performed (Supplementary Fig. 14). Furthermore, the thioflavin-T (ThT)-binding assay illustrated the formation of amyloid-like filaments under the combined action of Fmoc–F–F and His[24]. When 20 μM ThT was added to different sample solutions and excited at 438 nm, Fmoc–F–F (His) showed strong fluorescence absorption at 485 nm. The time course of ThT intensity during fibril formations was further tested. The fluorescence intensity of Fmoc–F–F in the presence of His showed an upward trend with longer co-assembly period, indicating that His could regulate the formation of fibril (Supplementary Fig. 15). In addition, zeta potential assay showed that Fmoc–F–F was ~–25 mV, while Fmoc–F–F (His) increased to ~+30 mV, proving the strong electrostatic interactions between His and Fmoc–F–F (Supplementary Fig. 16). Lastly, based on the small-angle X-ray scattering (SAXS) characterization of Fmoc–F–F (His) nanofilaments (Supplementary Fig. 17), the average radius ($R$) value at 8.18 nm was estimated by fitting the data using the cylinder model with SasView, which is consistent with the observation by Cryo-EM.

### Co-assembly mechanism of Fmoc–F–F (His)

To understand the co-assembly mechanism, experimental validation, and theoretical calculations were conducted. Firstly, the dynamic co-assembly process was demonstrated, as shown in Supplementary Fig. 18. Switching between filaments and nanorods was reversible, achieved by controlling the concentrations of Fmoc–F–F and His and

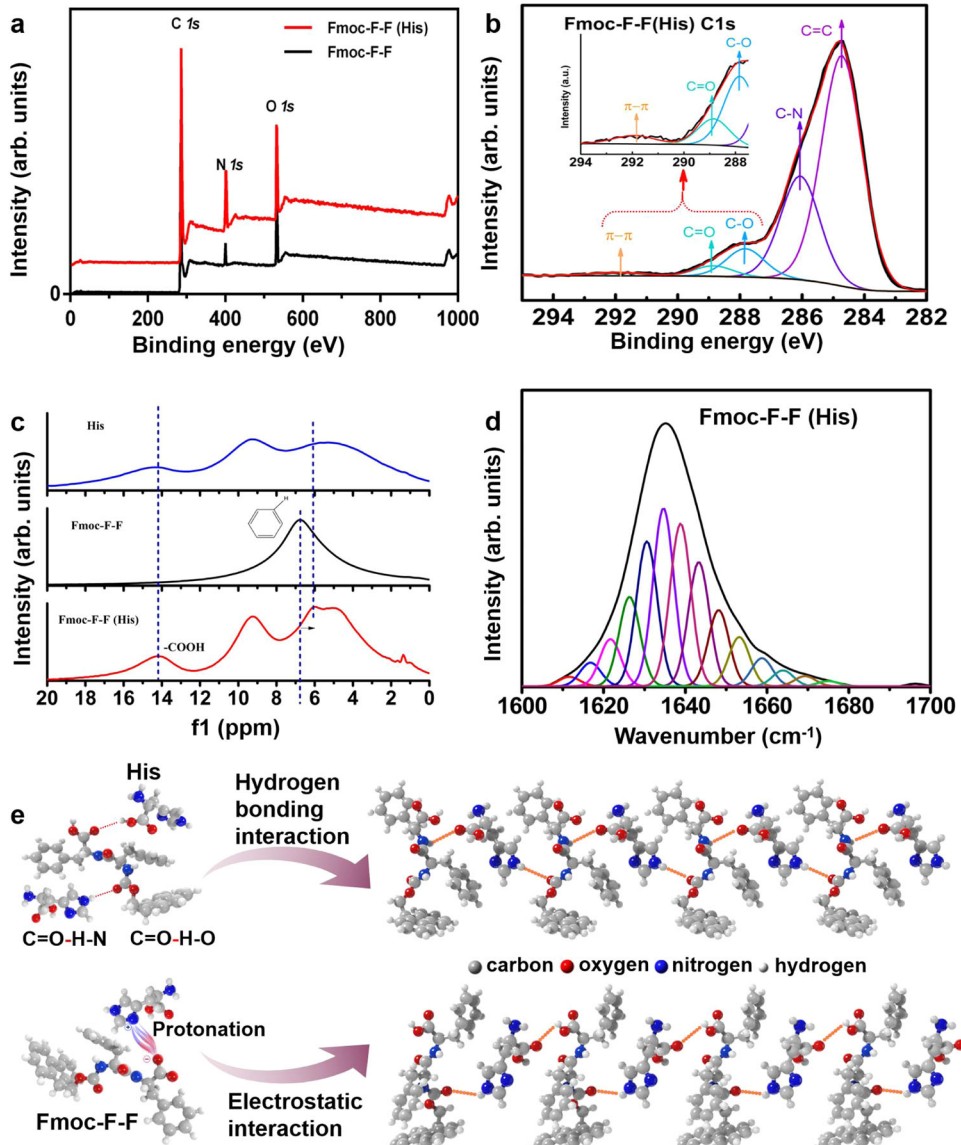

**Fig. 3 | Hydrogen bonding interactions of His and Fmoc–F–F destroy π–π stacking of F–F peptides and secondary structure. a** XPS of Fmoc–F–F (His) and Fmoc–F–F. **b** C *1s* peak of Fmoc–F–F (His) from XPS. **c** ¹H solid nuclear magnetic resonance (NMR) spectra of Fmoc–F–F (His) and Fmoc–F–F. **d** Percentage of secondary structure (Fmoc–F–F (His)) of amide I region (1600 cm⁻¹ to 1700 cm⁻¹) characterized by FTIR. **e** Possible interactions in the process of co-assembly between Fmoc–F–F and His. Representative images are shown. Source data are provided as a Source Data file.

disrupting the steady states of the co-assembly system. Secondly, X-ray photoelectron spectroscopy (XPS) showed that π–π accumulation in Fmoc–F–F was destroyed during co-assembly through C *1s* analysis of Fmoc–F–F and Fmoc–F–F (His) (Fig. 3a, b, Supplementary Fig. 19a and Table 1). ¹H NMR demonstrated that the chemical shift of C = C–H in Fmoc–F–F (His) was at 6 ppm, while the corresponding shit in Fmoc–F–F was at ~7 ppm, indicating a decrease of π–π accumulation in the presence of His (Fig. 3c). Lastly, FTIR of the stretching of the C = O groups in the amide I region was used to characterize secondary structures of the peptide assembly. Peak separation calculation was conducted through peak fitting of the amide I region, ranging from 1600 to 1700 cm⁻¹. The self-assembled Fmoc–F–F structure had a large number of antiparallel β-sheets (π–π stacking interaction) and a small number of α-helices (in the presence of –COO⁻) (The ¹H of –COOH of Fmoc–F–F at ~14 ppm was disappeared, shown in Fig. 3c) (Supplementary Figs. 19b and 20, Supplementary Table 3), which is consistent with the previous research[25]. However, these assembly modes were destroyed by His due to electrostatic interactions and hydrogen bonding and converted to β-sheets (Fig. 3d, Supplementary Table 2). According to the above characterizations, we provided a possible schematic of Fmoc–F–F (His) co-assembly to demonstrate the role of His in driving amyloid-like assembly of the nanofilaments of Fmoc–F–F by hydrogen bonding and electrostatic interactions (Fig. 3e).

To further understand the inhibitory mechanism of the Fmoc–F–F β-sheet-rich conformation through co-assembly with His, quantum

**Table 1 | Fmoc–F–F (His) and Fmoc–F–F XPS C *1s* peak splitting results statistics**

| Functional groups types | Fmoc–F–F (His) (%) | Fmoc–F–F (%) |
|---|---|---|
| C=C | 62.15 | 75.98 |
| C–N | 26.94 | 13.9 |
| C=O | 2.66 | 4.56 |
| π–π | 1.11 | 3.22 |
| C–O | 7.14 | 2.34 |

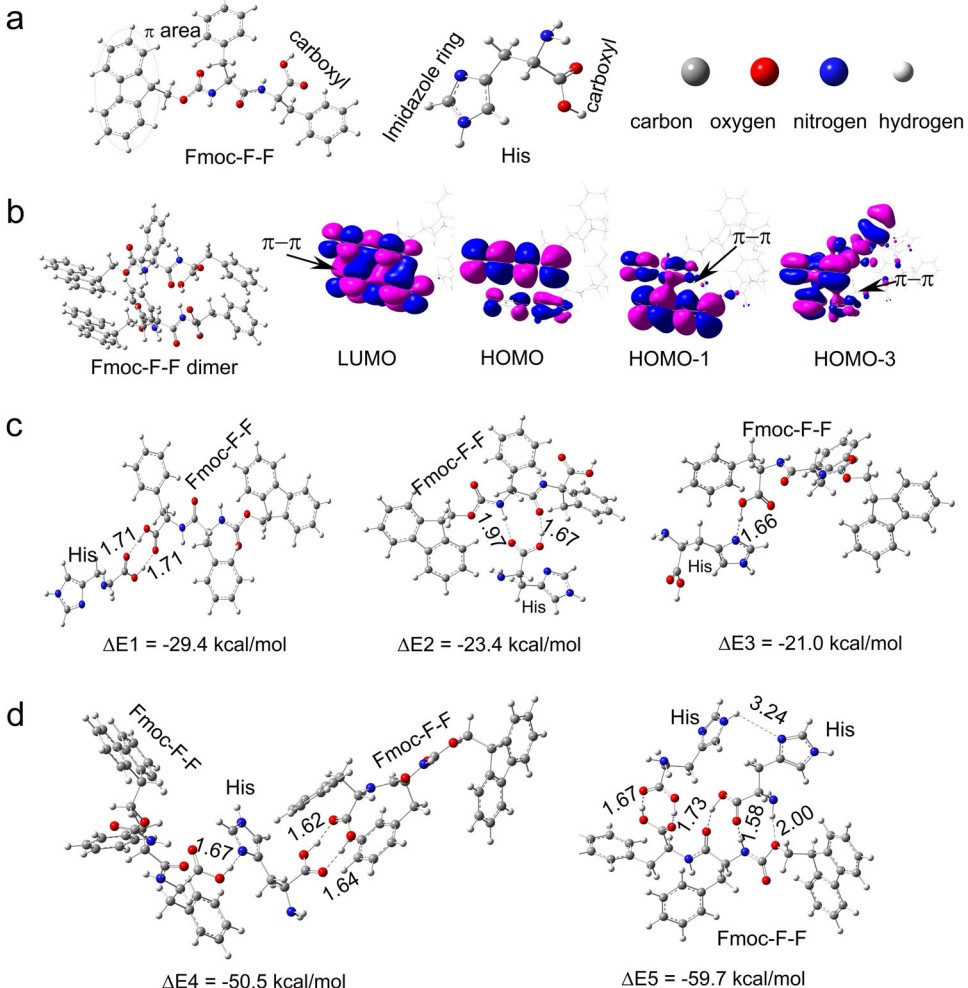

**Fig. 4 | Theoretical analyses of Fmoc–F–F and His co-assembly. a** Fmoc–F–F and His molecules. **b** Frontier orbitals of π-π stacking interaction of Fmoc–F–F dimer. **c** Multiple dimers of Fmoc–F–F and His are connected by different hydrogen bonding interaction modes. **d** Different trimers of 2Fmoc–F–F and His. Fmoc–F–F and 2His are connected by hydrogen bonding interaction modes. The atomic coordinates of the optimized computational models were provided in Supplementary Data 1.

calculations were conducted to assess the binding interactions between Fmoc–F–F and His molecules (Fig. 4a). The various Fmoc–F–F–His dimers were constructed and optimized using the density functional theory (DFT) calculations (Fig. 4c), and the results showed that these dimers were dominated by strong hydrogen bonding interactions, while the Fmoc–F–F–Fmoc–F–F dimers were dominated by π-π stacking interactions (Fig. 4b). The Fmoc–F–F–His dimers connected through double O–H···O–H hydrogen bonds between carboxyl groups in Fmoc–F–F and His showed the strongest binding energy. Of note, Fmoc–F–F molecules can connect two His molecules to form Fmoc–F–F–His·His trimers, and His molecules can connect two Fmoc–F–F molecules to form Fmoc–F–F·His·Fmoc–F–F trimers via different hydrogen bonding interactions (Fig. 4d, Supplementary Data 1). Thus, the molecular arrangements of Fmoc–F–F and His could be greatly influenced by the strong hydrogen bonding interactions between Fmoc–F–F and His, respectively, leading to further structural transitions. The experimental results indicated that an entangled network of Fmoc–F–F and His was observed in the presence of His, which may be attributed to the driving force in intermolecular interactions of hydrogen bonds, as shown in Fig. 2b. Furthermore, compared to those in self-assemblies with Fmoc–F–F alone, the classical β-sheet hydrogen bonds were prevented while the random hydrogen bonds were formed between Fmoc–F–F molecules in the co-

assemblies in the presence of His, leading to secondary structure transformation to form nanofilaments.

## Co-assembled Fmoc–F–F (His) nanofilaments exert enzyme-like activity to regulate ROS

Analysis showed that His conferred Fmoc–F–F with catalytic activity. The oxidoreductase and hydrolase activities of Fmoc–F–F (His) were monitored. Figure 5a demonstrated that Fmoc–F–F (His) exhibited high POD-like activity which catalyzes 3,3′,5,5′-tetramethylbenzidine (TMB) colorimetric reaction (the oxidized TMB with absorbance at 652 nm) in the presence of hydrogen peroxide ($H_2O_2$), but His exhibited low peroxidase (POD)-like activity and Fmoc–F–F exhibited almost no POD-like activity (Fig. 5a). In particular, the nanofilaments of Fmoc–F–F (His) remained intact structure with high catalytic activity after storage at room temperature for 60 days, showing a high stability (Supplementary Fig. 21). Importantly, the POD-like activity increased in Fmoc–F–F (His) assembly with longer term, indicating a positive correlation between catalysis and the assembly time (Supplementary Fig. 22). In contrast, neither Fmoc–F–F (His) nor Fmoc–F–F showed catalase (CAT)-like activity which catalyzes the decomposition of $H_2O_2$ into oxygen and $H_2O$, despite His showed high CAT-like activity (Fig. 5b). These results indicate that Fmoc–F–F (His) nanofilaments might be a nanozyme with POD-like activity which can be ascribed to

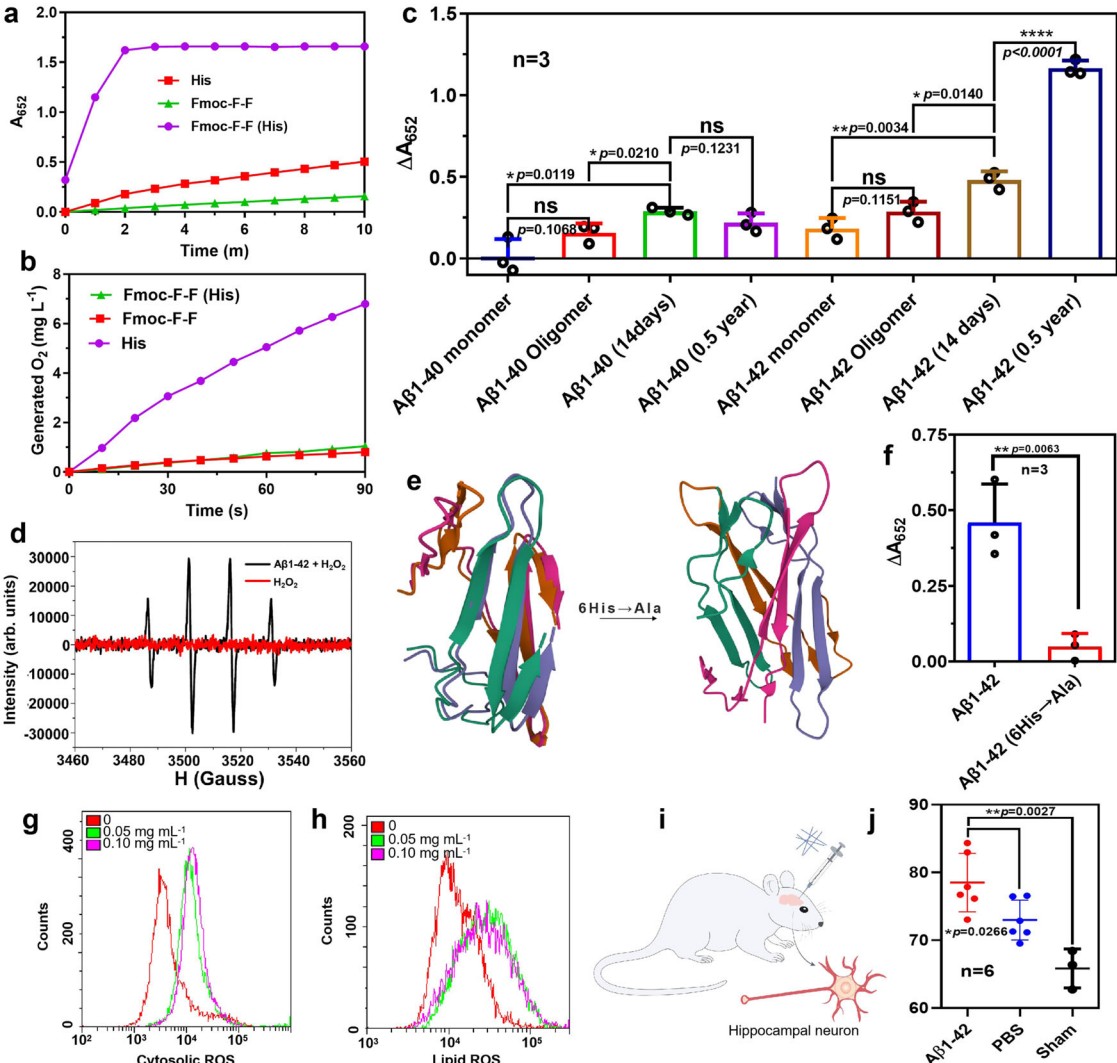

**Fig. 5 | Enzyme-like activity of Fmoc–F–F (His) and Aβ assembly. a** Peroxidase (POD)-like of Fmoc–F–F (His), Fmoc–F–F, and His, respectively. **b** Catalase (CAT)-like of Fmoc–F–F (His), Fmoc–F–F, and His, respectively. **c** POD-like activity of Aβ1–40 and Aβ1–42 filaments. The significant difference was evaluated by a two-tailed unpaired *t*-test. *n* = 3 independent samples, bars represent means ± SD, ns means no significance, ****$p < 0.0001$, **$p < 0.01$, *$p < 0.05$). **d** ESR characterization for •OH generation by POD-like activity of Aβ1–42 aggregates (half a year). **e** Predicted structures of Aβ1–42 and Aβ1–42 (6His→Ala) by alphafold2. **f** POD-like of Aβ1–42 and Aβ1–42 (6His→Ala). *n* = 3 independent samples. Mean ± SD is shown.

The significant difference was evaluated by a two-tailed unpaired *t*-test. **$p < 0.01$. Representative images are shown. **g** Cytosolic ROS (cROS) levels of HT-22 cells treated by different concentrations of Aβ1–42 filaments. **h** Lipid ROS levels of HT-22 cells treated by Aβ1–42 filaments. **i** Schematic diagram of injecting Aβ1–42 filaments into the hippocampus of SD rats. **j** Percentage of hippocampal histiocytes with high cROS level in SD rats after treatment of Aβ1–42 filaments. *n* = 6 biologically independent animals. Mean ± SD is shown. The significant difference was evaluated by a two-tailed unpaired *t*-test. **$p < 0.01$, *$p < 0.05$. Source data are provided as a Source Data file.

nanoscale assembly. The kinetics assays showed that. Fmoc–F–F (His) nanofilaments followed the typical Michaelis-Menten kinetics, in which the $K_M$ values (the Michaelis constant, representing the affinity of an enzyme to the substrate) of Fmoc–F–F (His) nanofilaments were 5.61 mM for $H_2O_2$ and 0.919 mM for TMB, respectively (Supplementary Fig. 23). Importantly, incubating Fmoc–F–F (His) nanofilaments with neuron cells significantly increased cellular ROS level (Supplementary Fig. 24).

Since Aβ polypeptide also contains F–F (19, 20) dipeptide and rich in His (6, 13, 14), it is hypothesized that its aggregates may perform similar catalytic properties due to the formation of nanofilaments. To prove our hypothesis, we prepared Aβ series aggregates (forms of Aβ shown in Supplementary Figs. 25–27) and checked their catalytic activity. We found that both Aβ1–40 and Aβ1–42 aggregates exhibited POD-like activities, in which Aβ1–42 aggregates showed higher activity than Aβ1–40 (Fig. 5c). In addition, Aβ incubated for 14 days (similar TEM to Fmoc–F–F) possessed higher POD-like activity compared with

the Aβ monomer and oligomers. When incubated for a longer time (half a year) at 37 °C, Aβ 1–42 formed filaments similar to Fmoc–F–F (His), and its catalytic activity was further enhanced. Similar to Fmoc–F–F (His) nanofilaments, the kinetics assays showed that Aβ 1–42 filaments also followed the Michaelis–Menten kinetics, in which the $K_M$ values were calculated to be 0.40 mM for $H_2O_2$ and 0.32 mM for TMB, respectively (Supplementary Fig. 28). To further confirm free radical generation, electron spin resonance (ESR) characterization was used to monitor the spectrum in the mixture of Aβ1–42 aggregates and $H_2O_2$. As shown in Fig. 5d, the featured peaks for hydroxyl radical (•OH) demonstrated that Aβ1–42 aggregates performed POD-like activity to generate ROS in the presence of $H_2O_2$).

To confirm the contribution of His to catalysis, the mutant with 6His replaced by Ala was prepared(Aβ1–42 (6His→Ala)), and the structure was predicted with Alphafold2 (Fig. 5e). In a tetramer assembly model, the secondary structure in wild-type Aβ1–42 is mainly β conformations (β sheet, β turn) and random coils. The

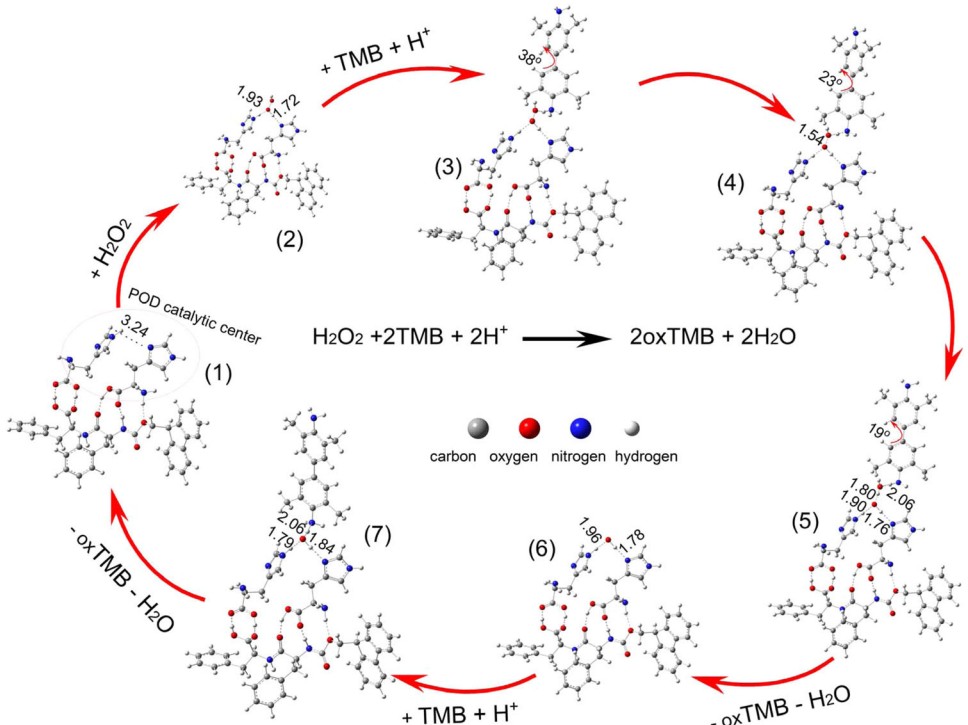

**Fig. 6 | Proposed mechanism of POD-like catalysis of Fmoc−F−F (His) and key structural parameters.** Seven reaction states mediated by POD-like catalysis of Fmoc−F−F (His) was proposed. The whole process can be divided into three major steps: The adsorption of $H_2O_2$ and ($H^+$ + TMB) molecules on Fmoc−F−F (His) (states 1−3); Oxidation reaction of the first TMB molecule oxidized by $H_2O_2^*$ under acidic conditions, producing $HO^*$, $H_2O^*$, and $oxTMB^*$ (states 4−5); Oxidation of second TMB molecule by second $HO^*$ under acidic conditions, producing $H_2O^*$ and $oxTMB^*$ (states 6−7). The corresponding binding energy of each step can be seen in Supplementary Fig. 33.

intermolecular distance between 6His and 19Phe−20Phe was 10.28 Å in tetramer model, which was much shorter than that of 13His and 14His (15.58 Å), indicating that 6His is more important for the construction of catalytic center of peroxidase-like activity (Supplementary Fig. 29). Such mutant Aβ1−42 assembly showed no POD-like activity compared to wild Aβ1−42 (Fig. 5f), indicating that POD activity of Aβ has strong relationship with His. In contrast, the predicted structure of the mutants of 13His→Ala and 14His→Ala showed no much influence on the distance between 6His and 19Phe−20Phe, indicating that both His may have less contribution on catalysis of Aβ1−42 assembly. However, if all three His were mutated (6, 13, 14His→Ala), Aβ1−42 would form a secondary structure mainly in random coils without β conformations, indicating that His is important to maintain β conformations. The cytotoxicity of Aβ 1-42 monomers, oligomers, and filaments were tested, and the result was shown in Supplementary Fig. 30. Their cytotoxicity (HT-22) may be related to their POD-like activity. Furthermore, Aβ1−42 filaments showed the ability to increase reactive oxygen species (ROS) levels (including lipid ROS, cytosolic ROS (cROS), and ROS of mitochondria) in HT-22 cells (hippocampus neuron[26]) (Fig. 5g, h and Supplementary Figs. 31 and 32). Consistently, the increased cROS level was observed when injecting Aβ 1−42 filaments into the bilateral hippocampus of Sprague Dawley rats (SD rats) (Fig. 5i), in which the percentage of cells with detectable cROS in hippocampal neurons significantly increased (Fig. 5j). These results indicate that Aβ1−42 filaments might be a ROS generator due to its POD activity.

To elucidate the POD-like mechanism of Fmoc−F−F (His), the Fmoc−F−F and two His molecule trimer was selected as the peroxidase catalytic reaction site (Fig. 6). We used DFT to calculate the reactants, products, intermediates, and transition states of the possible reaction pathway of the POD-like activities. As shown in Fig. 6, the imidazole rings of the two His molecules formed a catalytic center. The following

three reactions (Eqs. 1−3) serve as a plausible mechanism for the POD-like activities:

$$H_2O_2 + 2H^+ + 2TMB \rightarrow 2H_2O + 2oxTMB$$

$$H_2O_2 + (H^+ + TMB) + Fmoc - F - F (His) \rightarrow H_2O_2^* + (H_+ + TMB)^* \quad (1)$$

$$H_2O_2^* + (H^+ + TMB)^* \rightarrow HO^* + H_2O^* + oxTMB^* \quad (2)$$

$$HO^* + (H^+ + TMB)^* \rightarrow H_2O^* + oxTMB^* \quad (3)$$

Asterisk (*) marks species adsorbed on Fmoc−F−F (His). We then studied the POD-like activities of the oxidation reactions of the two TMB molecules oxidized by $H_2O_2$ under acidic conditions. The proposed reaction pathway is presented in Fig. 6, as well as the structural parameters for key intermediates in the POD-like catalytic cycle. The relative energies of the key intermediates are shown in Supplementary Fig. 33. As seen in Fig. 6; there were seven steps from the adsorption of $H_2O_2$ to the oxidation of the second TMB molecule, including six stable states and a transition state in the proposed reaction pathway. The dominant elementary reactions are presented in Eqs. 1−3, corresponding to the three steps in Supplementary Fig. 33, respectively: (1) adsorption of $H_2O_2$ and ($H^+$ + TMB) molecules on Fmoc−F−F (His); (2) oxidation reaction of the first TMB molecule oxidized by $H_2O_2^*$ under acidic conditions, producing $HO^*$, $H_2O^*$, and $oxTMB^*$; and (3) oxidation of second TMB molecule by second $HO^*$ under acidic conditions, producing $H_2O^*$ and $oxTMB^*$.

Our calculations indicated that the $H_2O_2$ molecule binds to the imidazole rings via hydrogen bonding with a binding energy of 0.63 eV

in the catalytic center, as shown in state (2) in Fig. 6. The hydrogen bonding interactions between $H_2O_2$* and imidazole rings facilitate the rapture of the O–O bond in $H_2O_2$*. The adsorbed $H_2O_2$ molecule in the catalytic center first breaks the O–O bond and one of the dissociated OH radicals binds to the H atom from TMB to form an $H_2O$* molecule and oxTMB* molecule. After the escape of $H_2O$* and oxTMB*, the remaining OH* radical in the catalytic center oxides the next TMB molecule. As shown in Supplementary Fig. 33, under acidic conditions, the first TMB molecule is easily oxidized by $H_2O_2$* with an energy barrier of 1.41 eV in step 2, while the second TMB molecule is easily oxidized by the OH* radical, as shown in step 3. From DFT analysis, the POD-like catalytic center of the co-assembly of Fmoc–F–F (His) effectively catalyzes the oxidation of TMB by $H_2O_2$ under acidic conditions due to the arrangement of His molecules in Fmoc–F–F (His) facilitating POD-like activity.

## Discussion

In this study, we demonstrated that His plays a critical role in modulating amyloid-like assembly and building active sites for Fmoc–F–F and Aβ aggregates. Distinct from the studies that most F–F assemblies have been conducted in organic solvents (e.g. DMSO, HFIP)[27,28], we demonstrated that the presence of an amino acid, particularly His, allowed Fmoc–F–F assembly from nanorods to nanofilaments under aqueous condition. In addition, the Fmoc–F–F (His)-assembled nanofilaments exhibit POD-like activity, demonstrating that His makes a major contribution to catalysis, which is similar to His residue in the active center of natural peroxidase[29]. These features demonstrate that F–F (His)-assembled nanofilaments are a type of POD-like nanozyme. Nanozymes refer, in particular, to the nanomaterials that can catalyze biochemical substrates of enzymes under physiological conditions, following similar enzymatic kinetics and thus performing enzyme-like catalysis, which has been recognized as next-generation artificial enzymes[30]. One of the major features is that nanozymes have multiple active sites on the surface, and thus the catalysis is determined by their nanostructures. Distinct from inorganic nanozymes, the Fmoc–F–F (His) nanofilaments performed POD-like activity depending on His mediated peptide assembled filament nanostructure, which contained a large number of active sites of His/F–F counterparts along the nanofilament. Therefore, Fmoc–F–F (His) nanofilaments are a nanozyme ascribed to their assembled nanostructure and catalysis, which is different from the structure–activity relationship of natural enzymes, although both of them are composed of amino acids.

In addition, Aβ1–42 filaments also exhibited POD-like activity, in which His may play an important role in the catalysis, as POD-like activity disappeared when 6His was mutated in Aβ1–42. These features demonstrated that Aβ1–42 filaments might also be a nanozyme. Importantly, Alphafold2 predication demonstrated that the active site of His/F–F may derive from intermolecular assembly, indicating that Aβ1–42 filaments are a catalytic unit with multiple active sites in the assembled format, which is distinct from traditional enzymes that often have single active center inside the protein framework. According to enzymatic kinetics assays, a single active site of His/F–F in Aβ1–42 may not be as efficient as that in natural peroxidase, which is composed of hemin coordinated with His residues[29]. However, one Aβ1–42 nanofilament contains multiple active sites in the assembled format, thus, the whole catalytic effect can not be negligible. In addition, the assembled Aβ1–42 nanofilaments are quite stable, hard to degrade, and can exist steadily for a long time, thus readily achieving durable catalytic activity. Therefore, different from traditional enzymes, Aβ1–42 filaments might act in the catalytic mode of nanozymes under a physiological environment.

Consequently, we speculated that Aβ1–42 filaments as a natural nanozyme with peroxidase-like activity may help understand the role of Aβ deposition in AD pathogenesis, as their detailed pathogenesis in AD is not yet fully elucidated. The Aβ cascade hypothesis is one of the most widely espoused theories of AD etiology, as Aβ aggregation and deposition exhibit strong associations with neurotoxicity and neurodegeneration[31]. In such a hypothesis, oxidative stress is considered a potential key pathway for Aβ to generate toxicity, which is usually ascribed to the aggregated form of Aβ coordinating with metal ions such as copper, iron, or zinc or enzyme cofactors such as heme. For example, positions 6, 13, and 14 of His can chelate copper ions, thereby generating ROS and causing neurotoxicity[32]. In addition, Aβ may bind to heme to form Aβ-heme complexes with POD activity, further generating ROS and neuronal damage[33]. However, we discovered that Aβ-aggregated filaments themselves exhibited POD-like activity and acted as an ROS initiator. In addition, the POD-like activity of Aβ1–42 filaments may be correlated with its aggregated form and deposition period. This phenomenon is consistent with the observation that when Aβ1–42 oligomers are converted to fibrils in plaques, the damage is increased and causes chronic neurodegeneration with subsequent cognitive impairment and dementia[34,35]. Thus, the discovery of intrinsic catalytic activity of Aβ aggregates may provide a new aspect of understanding the Aβ cascade hypothesis in AD etiology, i.e., assembled Aβ serves as a nanozyme to catalyze ROS generation and cause long-acting oxidative stress. In particular, it has been recently reported that Aβ can be easily aggregated in lysosomes[36], in which the acidic environment may profit them to perform POD-like activity, thus presenting a feasible way to generate ROS.

Overall, His modulation of Fmoc–F–F amyloid-like assembly and enhancement of catalytic activity demonstrated that introducing an amino acid into a peptide assembly is an effective strategy to program nanostructures and design peptide-based nanozymes. Aβ1–42 filaments were found to perform POD-like activity to enhance oxidative stress, which might also be ascribed to the interaction mode of His and F–F. Future work will focus on deciphering the precise structure of the assembled Fmoc–F–F (His) filaments and the role of His in Aβ assembly and catalysis, experimentally verifying whether His residues may be the targets for controlling Aβ aggregation and reducing neurotoxicity in AD.

## Methods
### Ethical regulations
All research complied with all relevant ethical regulations. All animal studies were performed following the protocols approved by the Institutional Animal Care and Use Committee of the Institute of Biophysics, Chinese Academy of Sciences.

### Reagents
The dipeptides and thioflavin-T (ThT) were purchased from Shanghai Yuanye Bio-Technology Co., Ltd. (China). Amino acids were purchased from Shanghai Macklin Biochemical Co., Ltd. (China). 3,3′,5,5′-Tetramethylbenzidine (TMB), 5,5-dimethyl-1-pyrroline N-oxide (DMPO), BODIPY 581/591 C11 probe and p-nitrophenyl acetate (pNPA) were purchased from Sigma-Aldrich Co., Ltd. (USA). M5 HiPure MitoSOX Red Mitochondrial Superoxide Indicator (MitoSOX probe) was purchased from Mei5 Biotechnology Co., Ltd. (China). DCFH-DA ROS detection kit was purchased from Beijing Fluorescence Biotechnology Co. Ltd. (China). The Acetylcholinesterase (AchE) Assay Kit was purchased from Beijing Solarbio Science & Technology Co., Ltd. (China). The Aβ peptides (purity > 95%) were purchased from GL Biochem (Shanghai, China).

### Preparation of dipeptide and amino acid assemblies
Using liquid chromatography-mass spectrometry (LC-MS), peptides with purities >97% were verified. The F–F was stored at −20 °C, and Fmoc–F–F was stored at −20 °C. For assembly, amino acids (concentration 20 mg mL⁻¹) were weighed and dissolved in deionized water after 8 min treatment of sonication. Peptides (concentration 2 mg mL⁻¹) were weighed and dispersed in amino acid aqueous

solution. After sonication treatment and standing a while (30 min), clarified Fmoc–F–F (His) instead of cloudy Fmoc–F–F solution was prepared.

## Electron microscopy for peptide samples
Samples were dropped on a silicon wafer at room temperature (-23 °C). The SEM images were recorded using a Hitachi SU8010 (Japan), and TEM images were recorded using a Tecnai Spirit (FEI) (USA) operating at 100 kV dropped on a double copper net. For AFM, liquid Fmoc–F–F (His) (1 mL) was dropped on a mica sheet and observed by Bruker Dension Icon (Germany). For negative staining TEM, the co-assembled Fmoc–F–F (His) was stained with uranyl acetate. Cryo-EM and two-dimensional classification data were collected by Talos Arctica 200 kV FEG (USA), and the sample was pre-frozen at −184 °C.

## Characterizations based on spectroscopy, XRD, NMR, SAXS, ThT binding, and zeta potential
The samples were characterized using FTIR spectroscopy. For FTIR, 3 mg of powdered sample and 50 mg of potassium bromide (KBr) were mixed for tablet compression analysis with transmittance at 400–4000 $cm^{-1}$, collected using an iS10 FTIR spectrometer (USA). Measurements were performed by averaging 32 scans at 4-$cm^{-1}$ resolution (signal-to-noise ratio (S/N) = 50000). The samples were characterized by XRD using a D8 ADVANCE (Germany). Measurements were performed at 1.5406 angstroms of the copper target wavelength, 40-kV tube voltage, and 40-mA tube current. As for NMR, solid samples were characterized by JNM-ECZ600R at a frequency of 600 MHz at room temperature. The tube diameter, mass frequency, and relaxation delay were set as 3.2 mm, 15 kHz, and 5 s by averaging 16 scans.

SAXS measurements were performed on the solution samples of a concentration of 3.75 mM loaded in a low noise flow cell using a Xenocs Xeuss 2.0 instrument equipped with a Dectris Pilatus 300k detector (pixel size 172 μm) and a Cu-Kα radiation source (wavelength $\lambda$ = 1.54189 Å). 2D SAXS data were collected at an exposure time of 1200 s (room temperature). The SAXS data of the Silver Behenate standard were also collected and used to calibrate the sample-to-detector distance (2480 mm). The 2D SAXS data of the samples were reduced to obtain 1D scattering intensity profile $I(q)$, where $q$ is the scattering vector ($q = 4\pi \sin\theta/\lambda$ (4), $2\theta$ is the scattering angle). The raw data were corrected for the sample background (flow cell containing water) in the data reduction process using the SasView5.0.6 software package[37,38] (the instructions could be obtained from https://www.sasview.org/).

For ThT-binding analysis, 20 μL of assembled solution was added to 160 μL of $DDH_2O$ in a 96-well blackboard. Then, 20 μL of 200 μM ThT was immediately added. Excitation was set to 438 nm, and fluorescence intensity was recorded at an emission wavelength of 485 nm.

Zeta potentials of samples were determined using a Zetasizer Nano ZS90 (England) via three tests in a 1-mL sample cell. The UV spectrum at 200–800 nm was collected using a Tanon-3500R Gel Imaging System (Shanghai, China). The fluorescence emission spectrum at 300–550 nm and 280 nm excitation was collected using a F-7000 FL spectrophotometer (Japan) with a scan speed of 1200 nm/min. The PMT voltage and response were set to 500 V and 0.1 s, respectively. Both EX Slit and EM Slit were set to 2.5 nm.

## Assays for enzyme-like activity
For POD activity, the reaction system contained 700 μL of sodium acetate buffer (0.1 M, pH 3.5), 100 μL of TMB (2.4 mg mL$^{-1}$ in DMSO), 100 μL of Fmoc–F–F (His), and 100 μL of 30% $H_2O_2$ (10 M) in a 4-mL optical path 1-cm glass cuvette. After quickly mixing the components, absorbance at 652 nm was monitored with a UV–VIS spectrophotometer (China) at room temperature. To detect Aβ activity, 140 μL

of buffer, 20 μL of TMB, 20 μL of Aβ, and 20 μL of $H_2O_2$ (625 mM) were added in order and reacted for 2 h at 37 °C. Absorbance at 652 nm was monitored using a microplate reader. For CAT activity, 500 mM $H_2O_2$ solution was prepared, with 3.6 mL of $H_2O_2$ and 0.4 mL of Fmoc–F–F (His) added. Immediately, the COD analyzer probe was inserted into the reaction solution to record the number of different reaction times. Evaluation of •OH was performed by the interaction between Aβ and $H_2O_2$. Briefly, DMPO (50 mM) and $H_2O_2$ (62.5 mM) were dissolved in sodium acetate buffer (0.1 M, pH 3.5) containing Aβ (100 μg mL$^{-1}$). The product DMPO-OH was detected using an A300-10/12 ESR instrument (Germany).

## Computational methods
All quantum chemical calculations were performed using DFT methods with an empirical dispersion correction (D3) implemented using the Gaussian09 package. Typical Fmoc–F–F/His dimers and trimers predominantly stabilized by the intermolecular hydrogen bonds were constructed and optimized at the level of B3LYP/6-31 g (d) theory. Harmonic vibrational frequency calculations of the optimized geometries were also performed to ensure the structures at local minima. The Fmoc–F–F/His binding energies ($\Delta E$) were calculated as the energy difference between their molecular clusters and the sum of the energies of Fmoc–F–F and His (Eq. 4).

$$\Delta E = E_{(Fmoc-F-F)m(His)n} - mE_{Fmoc-F-F} - nE_{His} \qquad (4)$$

## Cellular and in vivo ROS assessment
HT-22 cells (SCC129, Sigma Aldrich/Merck) were maintained in RPMI 1640 medium supplemented with 10% fetal bovine serum (FBS) and 1% penicillin and streptomycin (PS). $1 \times 10^5$ cells per well were plated in 12-well plates and cultured with 5% $CO_2$ at 37 °C for 24 h. HT-22 was treated with different concentrations of Aβ1–42 filaments for 24 h. For oxidative stress assessment, cells were incubated with 200 μL of 2 μM BODIPY 581/591 C11 probe, 4 μM MitoSOX probe, or 10 μM DCFH-DA working solution for 30 min at 37 °C (except MitoSOX probe for 10 min). To measure intracellular lipid peroxidation, ROS of Mitochondria, and cytosolic ROS (cROS), respectively. After digestion with trypsin, $1 \times 10^4$ cells were collected and analyzed using flow cytometry (BD FACSCalibur) to measure the fluorescence intensity of these ROS levels. For cytotoxicity assay, HT-22 cells were planted in 96-well plates ($6 \times 10^3$ cells per well) and cultured with 5% $CO_2$ at 37 °C overnight. Then the culture medium was replaced by fresh DMEM medium with 10% FBS and 1% PS, 100 μg mL$^{-1}$ of Aβ 1–42 monomer, oligomer, and filament were added to incubate for another 24 h. Then the medium containing materials was removed, and cell viability was measured using a CCK8 Kit. The cROS detection in PC-12 cells (CRL-1721, ATCC) (They were maintained in RPMI 1640 medium supplemented with 10% horse serum (HS), FBS 5% FBS, 1% PS, and pre-induced with 50 ng mL$^{-1}$ nerve growth factor (NGF) for neuronal differentiation for 10 d) was detected in adding 10 μM DCFH-DA working solution for 30 min at 37 °C. After digestion with trypsin, $1 \times 10^4$ cells were collected and analyzed using flow cytometry (BD FACSCalibur) (following the gating strategy in Supplementary Fig. 31) for measuring the fluorescence intensity of these cROS levels.

To evaluate the ability of Aβ 1–42 filaments to generate cROS in vivo, 10 μl Aβ filaments (1 mg mL$^{-1}$) and PBS were injected into the bilateral hippocampus of the male SD rats[39] (6–8 weeks, all animals were housed in a temperature controlled (25 °C) condition, and were given free access to water and diet.), respectively. After the SD rats were observed for 5 days, the right hippocampus tissue was taken and prepared into a cell suspension following the below procedures. Firstly, the rats were dislocated and sacrificed after anesthesia, and the brain was collected (about 3–5 min). Next, after washing with normal saline, the right hippocampus was peeled off on the ice bag and quickly

immersed in a dish containing PBS solution (2 min). All the hippocampus samples were then transferred into a 1.5 ml PE tube and incubated with trypsin (0.25%, 1 mL) for digestion in a water bath at 37 °C for 10 min. The serum was subsequently added to stop the digestion, and the hippocampus cells were collected by filtration with 300 micron nylon mesh and centrifugation at 300×*g* for 5 min. Then the cells were resuspended with PBS and stained with DCFH-DA following the instructions of the ROS detection kit (Beijing Fluorescence Biotechnology Co. Ltd., China). The cytosolic ROS was detected using flow cytometry. Only cells in normal size were selected for analysis during flow loop gate[40].

### Statistics and reproducibility

Three times each experiment was repeated independently with similar results. The significance of the data in Fig. 5c, f, j was analyzed according to unpaired Student's two-sided *t*-test by GraphPad Prism 7.0. *ns* means no significance, $*p < 0.05$ $**p < 0.01$ and $****p < 0.0001$.

### Reporting summary

Further information on research design is available in the Nature Portfolio Reporting Summary linked to this article.

## Data availability

All data are available in the main text or Supplementary Materials. All electron microscope images are available upon request to the corresponding authors. Source data are provided in this paper. The optimized atomic coordinates of the Fmoc−F−F molecule are available as Supplementary Data 1. Source data are provided in this paper.

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

## Acknowledgements

We thank Dr. Jianping Jia from the Clinical Center for Neurodegenerative Disease and Memory Impairment, Capital Medical University, for the discussion of the relationship between Aβ filament catalysis and AD pathogenesis. We thank the Testing and Analysis Center at the Institute of Biophysics, Chinese Academy of Sciences (CAS), for the characterization of nanomaterials. We thank Can Peng for help with SEM samples. We thank Shuoguo Li and Yun Feng from the Center for Biological Imaging (CBI), Institute of Biophysics, CAS, for help with taking and analyzing TEM, Prof. Ningdong Huang from the University of Science and Technology of China for help with SAXS analysis. This work was supported by the National Key R&D Program of China (2019YFA0709200), the National Natural Science Foundation of China (32201162), the 70th general grant of China Postdoctoral Science Foundation (2021M702947), National Natural Science Foundation of China Foundation of Innovative Research Group grant (22121003), and National Natural Science Foundation of China (81930050).

## Author contributions

L.G. conceived and organized the project. Y.Y. participated in all experiments. L.C. performed theoretical calculations and participated in structural and logical analyses. L.K. performed and collected cryo-EM, negative staining TEM, and two-dimensional classification data. L.Q., M.C., Z.F., S.M., M.S., C.Z., and X.W. performed the cell tests and data analysis. J.J. guided the experiments. Y.L. performed TEM. L.G., X.Z., and L.W. contributed to the data discussion and drafted the paper. All authors contributed to the editing of the paper.

## Competing interests

The authors declare no competing interests.
