## [Peer Review File · Nature Communications]

REVIEWER COMMENTS

Reviewer #1 (Remarks to the Author):

Comments

Title: Histidine modulates dipeptide assembly of A β motif and confers 2 enzyme-like activity

Ye Yuan et al. confirmed that histidine could facilitate assembly of A β series peptides and confer A β assembly as a nanozyme to generate reactive oxygen species (ROS) in AD pathogenesis. In the present study, the authors investigated the co-assembly behaviour of the Fmoc-F-F dipeptide in an aqueous solution containing 20 natural amino acids. They confirmed that histidine (His) depolymerizes the π - π stacking of Fmoc-F-F and converted the assembly of dipeptides from nanorods into nanofilaments. Overall, the methodology applied in this paper seems to be reasonable and the results are correlated well with the previous literature.

Hence, I firmly accept this manuscript for publication. However, some important points should be clarified before publication.

1. The authors optimized the Fmoc-F-F-His dimers (Fig. 4a) using DFT calculations. Authors are requested to compare the simulated IR and Raman spectra to the observed FT-IR and Raman spectra (Refer. Journal of Molecular Structure, 1213 (2020)).

2. Add some references supporting the Raman and Infrared spectral analysis results.

Journal of Molecular Structure, 1213 (2020) 128163; Journal of Molecular Liquids, 290 (2019) 111209; Spectrochimica Acta Part A, 105 (2013) 218-222.

3. Add some references related to the title compound in the introduction part.

Chem. Rev. 2021, 121, 22, 13869–13914; Spectroscopy letters 47 642–648 (2014).

Reviewer #2 (Remarks to the Author):

The manuscript by Ye Yuan et al., describes the characterisation of the FMOC-FF with and without the addition of Histidine amino acid. They also explore the effect of other amino acids on morphology. The authors then characterise the ability of the mixture to catalyse a reaction. Finally, the author examine the catalytic ability of Amyloid-beta 1-40 and 1-42.

Overall this is a thorough and extensive investigation of the phenomenon of mixing FMOC-FF with His and the outcome of this. The authors suggest that this is a co-assembly although it remains uncertain whether they are able to distinguish between self-sorting and co-assembly. What is clear though is that the HIS has an influence on morphology of the FMOC-FF and that the presence of HIS confers some ability to generate ROS. It appears that His alone is quite active in generating Oxygen.

The authors then move on to the separate and unrelated peptide, Amyloid-beta (1-40 and 1-42) to investigate whether it is also able to catalyse the reaction. The generation of ROS by Amyloid-beta has been shown before.

The experimental work is well done. The catalytic assays are suggestive of catalysis but the thorough enzyme kinetics analysis is missing. The paper includes very nice CryoEM images but analysis of the data is missing.

I have a number of comments that must be corrected and addressed

1) FMOC-FF is not anything like Amyloid-beta. It does not mimic Abeta and shares very little sequence similarity (two amino acids). Amyloid beta does not possess the large FMOC and I do not believe that the authors can claim similarity between a small molecule dipeptide with FMOC and a 42 amino acid protein. Therefore, this needs editing throughout. There is no evidence that FMOC-FF is able to self-assemble in any similar way to the large peptide. There is no evidence that FMOC-FF is able to give any information about the behaviour of Abeta 1-42 or to Alzheimer's disease mechanisms.

therefore, the paper should be edited. The FMOC-FF section should separate from the Abeta section.

e.g.

The first sentence of abstract is incorrect. How is a FMOC-FF small molecule "derived" from Amyloid beta. Many proteins have FF present. We can not claim that FF "derives" from these either.

"Alzheimer's disease (AD) amyloid-beta (A β) polypeptide-derived diphenylalanine peptide (FF)" - remove.

Also from the title, which must be corrected

"dipeptide assembly of A β motif"

"and may not fully mimic the A β assemblies that occurs in the physiological environment".

I would suggest it doesn't mimic it at all and this should be removed

2) Fibrosis is not the correct word to use. Fibrillogenesis is preferable. Fibrosis means something entirely different. This is found both in the main text and also in the Suppl. Throughout the text requires careful proof reading including in the abstract. e.g longer time aggregation etc.

3) I think the authors mean sonication, rather than "ultra-sonic process"?

4) Can a three component (FMOC and two amino acids) really be claimed to form a-helical or beta sheet structures? Given the expected 3.6 residues per turn for an alpha helix? I don't think this is reasonable to claim this small molecule forms protein-like conformations.

5)"involuntarily" should be replaced with an alternative. This implies intention and doesn't make sense in this context.

6)When the authors say Cystine, they mean cysteine (the amino acid) not the disulphide bonded form which is cystine.

7)the authors showed cryo EM showing a regular fibrillar structure - could they have done more analysis on these structures to gain atomic information on the arrangements. What is the pitch? What is the significance of the measurements? these have not been put into context of the structural organisation of the small molecule.

8) Page 7 we are told "Third, Fmoc-F-F (His) crystal peak diffraction intensity was high, sharp, and scattered, indicating a good crystal state. In addition, the fewer peaks of Fmoc-F-F (His) accompanied by a shift in the diffraction peaks suggested that His may enter the original crystal lattice and cause distortion. Strong diffraction peaks at 21° and 24° were formed (Supplementary Fig. 13)."

But there is no intro to this technique - is this single crystal X-ray crystallography? No Data is shown. Is this fibre diffraction? More information is required here to make sense of this information

9) p11. what is DFT?

10) Enzyme kinetics is not shown and I think this is needed to clearly show catalysis

11) I could not find any information about the relative amounts of FMOC-FF and His? What is the ratio?

12) Figure showing Alpha-fold prediction needs further explanation. If the structures with His6Ala is different, it could be the reason for different activities (rather than missing His). Plus there are two additional HIS residues within Abeta. Why are these not considered - especially since they are consecutive, His-His?

12) Supplementary section requires careful proof reading throughout

Reviewer #3 (Remarks to the Author):

In this study, the authors used Fmoc-F-F peptides with/without His to study their propensity to form fibrillar aggregates. They found that His has a property to modulate the physical properties of Fmoc-F-F dipeptides. Next, inspired by this findings, they studied the possible POD-like activity of Ab1-40 and Ab1-42 because the F-F motif is seen in the amyloid fibrils made from these proteins. They successfully discovered that Ab1-40 and Ab1-42 fibrils show POD-like activity, thereby producing ROS species. They discuss that the enzymatic activity of Ab fibrils may be related to the pathogenesis of AD.

The findings presented in this paper will shed new light on the molecular mechanism of the pathogenesis of AD. To the reviewer's knowledge, this is the first study to discover that amyloid fibrils show enzymatic activity. This work will play an important role in elucidating the mechanism of amyloidosis.

On other other hand, there are some points that the authors should focus on. I will describe these points below. In particular, analysis and interpretation of small-angle X-ray scattering data should be done correctly.

Major:

Both Fmoc-F-F and Fmoc-F-F(His) increase ThT fluorescence intensity, suggesting that both forms amyloid-like structures. To better understand the mechanism of fibril formations, the authors should study the time-course of ThT intensity changes during fibril formations for both samples.

Relating to this, to obtain more consistent results with measurements using Ab, POD-activity of intermediate oligomer states of Fmoc-F-F and Fmoc-F-F(His) should be investigated. This will clarify whether the POD-activity of the minimal motif is indeed closely related to the POD-activity of Ab proteins because the authors already detected that Ab fibril POD-activity is stronger than Ab monomers and oligomers.

II. 208-211 Treatment of the SAXS data is wrong. The value of -0.03758 means nothing. Cross-sectional Guinier plots or Guinier plots should be applied to this data first. Then, the cross-sectional radius of gyration (R_c) or the radius of gyration (R_g) values should be estimated.

Further, the equation used for (cross-sectional) Guinier plots should be described.

Supplementary Fig. 19

The SAXS curve should be displayed in the logarithmic scale. In the inset, the scale of the ordinate should be changed such that the curve is more clearly seen. Both the measured data points and the fit should be displayed. The abscissa should start from $q^2=0.0$ [nm⁻²]

II. 287-290 The authors found that Ab fibrils have higher POD-like activity compared with monomers and oligomers. Does this mean that fibrils are more cytotoxic than oligomers? The authors should discuss this point.

II. 420-421 Sample concentrations, quantities, exposure times, and how the data reduction was conducted should be written here for SAXS measurements.

Minor:

II. 34-37 This sentence can read as if Ab causes Parkinson's disease as well as AD.

Is it true?

Reviewer #4 (Remarks to the Author):

This is a thorough and interesting study, combining experiment and molecular that deserves publication. The extrapolation of the properties of the FF assembly structure to Abeta assembly and neurodegenerative diseases is a bit far fetch. This said I think the results in itself is an interesting example of nano-construct that can have a catalytic activity. It should be termed catalytic activity not

enzymatic activity as it lacks the specificity and selectivity of a full-fledged enzyme. I think the term nano-enzyme is very misleading.

I do not think the results are general and urgent enough to be published in Nature Communication. I would suggest publication in a more physical chemistry oriented journal. For such a publication it would be interesting to investigate the influence of ionic strength and buffer composition on the formed nano-structures. Such information is crucial in order to assay the applicability of the formed nano-structures. It would also give further insight on the forces that regulates the assembly process.

Reviewer #5 (Remarks to the Author):

Comments of Referee

In the biochemical and cell culture assays of the study, the authors investigated the effects of A β peptides (A β 1-40 and A β 1-42) on oxidative stress and reactive oxygen species (ROS) productions in the mouse hippocampal (HT-22) cell line and bilateral hippocampus of SD rats. The generation of ROS was increased in the cell line and rat hippocampus by increasing the POD activity, although the activity of catalase was decreased. The authors observed that A β 1-42 filaments induced ROS generation actions due to their POD activities. The study is potentially interesting for the journal, and it has also high citation potential. However, it is careless presentation. There seem to be areas of considerable scientific confusion in the sections of methods, results, and discussion, especially. Before making a judgment regarding the document, the author needs to respond appropriately to the main criticisms listed below.

Major remarks

1. In 'Cellular and in vivo ROS assessment'. The section is too short. For example, how many cells such as 1×10^6 and 2×10^6 cell were used for the analyses? Were the HT-22 cells obtained? For the dose of $10 \mu\text{l}$ A β filaments (1mg mL^{-1}), please add a reference or reason. For obtaining the hippocampal neurons, please add a reference or reason. There are no details of gender and age of animals in the section.
2. Mitochondria have great importance on the generation of ROS in the cells and neurons. The detection of mitochondrial ROS by using a probe MitoSOX may be useful for the current study?
3. Discussion section is too short and it lacks focus. The author should concentrate on interpretation of their findings and their relevance to the field of study. The discussion should be used for the interpretation of data and for pointing out the significance of the findings.
4. There are serious problems in the abbreviations of the manuscript. The authors used both abbreviated and full names of several words such as ROS and Histidine through the manuscript.
5. DCFH-DA is a marker of cytosolic ROS. Please use 'cytosolic ROS (cROS)' instead of 'ROS' through the manuscript.
6. DCFH-DA is a fluorogenic probe that measures hydroxyl, peroxy and other ROS activity within the cell. The authors prepared hippocampus homogenate without collagenase and other procedures. The DCFH-

DA is not suitable for the cell homogenate (injured cells), although it is suitable for HT-22 and whole hippocampal neurons. The authors should give detailed information on the preparation of hippocampal samples in the section.

In abstract

- In '...correlation with Histidine'. Please use 'His' instead of 'Histidine'.
- Please delete once used ROS abbreviation in the 'generate reactive oxygen species (ROS)'.

In results

- In 'SD Rats', 'Co-assembled Fmoc-F-F (His) nanofilaments exert enzyme-like activity to regulate ROS'. Please use the full name of the SD abbreviation before using it through the manuscript.
- In page 10, ... high POD-like activity which catalyzes TMB colorimetric reaction (the oxidized TMB with absorbance at 652 nm)... Please use 3,3',5,5'-tetramethylbenzidine (TMB) instead of 'TMB'.
- In page 11,the oxidation reactions of the two 3,3',5,5'-tetramethylbenzidine (TMB) molecules.. Please use 'TMB' instead of 'two 3,3',5,5'-tetramethylbenzidine (TMB)'.
- Please add p values to the section.

In material and methods

- 'In 'Cellular and in vivo ROS assessment'. Please use 'ROS detection kit' instead of 'reactive oxygen species detection kit'.

In figure legends

- In Fig. 5. The unit of ROS and lipid ROS are different from the reference values. For ROS fluorescence intensity unit, arbitrary unit may be useful.

Histidine modulates dipeptide assembly of A β motif and confers enzyme-like activity

Comments of Referee

In the biochemical and cell culture assays of the study, the authors investigated effects of A β peptides (A β 1-40 and A β 1-42) on reactive oxygen species (ROS) production in the mouse hippocampal (HT-22) cell line and bilateral hippocampus of SD rats. The generation of ROS was increased in the cell line and rat hippocampus by increasing the POD activity. The authors observed that A β 1-42 filaments induced a ROS generation action due to its POD activity. The study is potentially interesting for the journal, and it has also high citation potential. However, it is careless presentation. There seem to be areas of considerable scientific confusion in the sections of methods, results, and discussion, especially. Before decision on the manuscript, the author should give appropriate answer to major comments at the below.

Major remarks

1. In 'Cellular and *in vivo* ROS assessment'. The section is too short. For example, how many cells such as 1×10^6 and 2×10^6 cell were used for the analyses? Were the HT-22 cells obtained? For the dose of $10 \mu\text{l}$ A β filaments (1mg mL^{-1}), please add a reference or reason. For obtaining the hippocampal neurons, please add a reference or reason. There are no details of gender and age of animals in the section.
2. Mitochondria have great importance on the generation of ROS in the cells and neurons. The detection of mitochondrial ROS by using a probe MitoSOX may be useful for the current study?
3. Discussion section is too short and it lacks focus. The author should concentrate on interpretation of their findings and their relevance to the field of study. The discussion should be used for the interpretation of data and for pointing out the significance of the findings.
4. There are serious problems in the abbreviations of the manuscript. The authors used both abbreviated and full names of several words such as ROS and Histidine through the manuscript.
5. DCFH-DA is a marker of cytosolic ROS. Please use 'cytosolic ROS (cROS)' instead of 'ROS' through the manuscript.
6. DCFH-DA is a fluorogenic probe that measures hydroxyl, peroxy and other ROS activity within the cell. The authors prepared hippocampus homogenate without collagenase and other procedures. The DCFH-DA is not suitable for the cell homogenate (injured cells), although it is suitable for HT-22 and whole hippocampal neurons. The authors should give detailed information on the preparation of hippocampal samples in the section.

In abstract

- In '...correlation with Histidine'. Please use 'His' instead of 'Histidine'.
- Please delete once used ROS abbreviation in the 'generate reactive oxygen species (ROS)'.

In results

- In 'SD Rats', 'Co-assembled Fmoc-F-F (His) nanofilaments exert enzyme-like activity to regulate ROS'. Please use the full name of the SD abbreviation before using it through the manuscript.
- In page 10, ... high POD-like activity which catalyzes TMB colorimetric reaction (the oxidized TMB with absorbance at 652 nm)... Please use 3,3',5,5'-tetramethylbenzidine (TMB) instead of 'TMB'.
- In page 11, ...the oxidation reactions of the two 3,3',5,5'-tetramethylbenzidine (TMB) molecules.. Please use 'TMB' instead of 'two 3,3',5,5'-tetramethylbenzidine (TMB)'.

- Please add p values to the section.

In material and methods

- 'In 'Cellular and *in vivo* ROS assessment'. Please use 'ROS detection kit' instead of 'reactive oxygen species detection kit'.

In figure legends

- In Fig. 5. The unit of ROS and lipid ROS are different from the reference values. For ROS fluorescence intensity unit, arbitrary unit may be useful.

The point-to-point response has been prepared as the followings:

Review 1:

Ye Yuan et al. confirmed that histidine could facilitate assembly of A β series peptides and confer A β assembly as a nanozyme to generate reactive oxygen species (ROS) in AD pathogenesis. In the present study, the authors investigated the co-assembly behaviour of the Fmoc-F-F dipeptide in an aqueous solution containing 20 natural amino acids. They confirmed that histidine (His) depolymerizes the π - π stacking of Fmoc-F-F and converted the assembly of dipeptides from nanorods into nanofilaments. Overall, the methodology applied in this paper seems to be reasonable and the results are correlated well with the previous literature. Hence, I firmly accept this manuscript for publication. However, some important points should be clarified before publication.

Response: We really appreciate the reviewer's positive comments.

1. The authors optimized the Fmoc-F-F-His dimers (Fig. 4a) using DFT calculations. Authors are requested to compare the simulated IR and Raman spectra to the observed FT-IR and Raman spectra (Refer. Journal of Molecular Structure, 1213 (2020)).

Response: We are thankful to the referee for the constructive comments. In the present study, the structural and spectroscopic investigations of His, Fmoc-F-F and Fmoc-F-F (His) co-assembly were performed using density functional theory (DFT) quantum chemical calculations and validated by experimentally. The molecular structure of the molecule was optimized and the harmonic vibrational wavenumbers of the molecule were calculated. The vibrational wavenumbers were also observed using FTIR and Raman spectroscopy. The calculated and observed vibrational wavenumbers were assigned.

Figure for review 1 The simulated (black curves) and observed (red curves) Raman spectra of the Fmoc-F-F molecule.

According to the reviewer's suggestion, first, the Raman spectrum of the Fmoc-F-F molecule (Fmoc-F-F_DFT) was simulated using DFT calculations and presented in Figure for review 1 together with the observed Raman spectrum (Fmoc-F-F_Exp), which indicates that the calculated Raman vibrations of the Fmoc-F-F molecule in the range of 1000 - 1600 cm^{-1} and 2800 - 3200 cm^{-1} are in good agreement with that of the experimental results. There are 4 benzene rings in a Fmoc-F-F molecule, and the vibration behaviors of benzene ring are the main components in the FTIR and Raman spectra. The C=C and C-C stretching vibrations of benzene ring are very much prominent and highly characteristic vibrations in vibration spectra of benzene ring and its derivatives. The benzene ring C=C and C-C stretching vibrational modes usually appear in the region 1650 - 1200 cm^{-1} . (*Journal of Molecular Structure* 1213 (2020) 128163) The calculated ring C=C stretching vibrations of the Fmoc-F-F molecule falls in the frequency region 1700 - 1400 cm^{-1} . The strong peaks at 1470 cm^{-1} and 1700 cm^{-1} in Raman spectra were assigned to couple vibrations of C=N and C=C stretching and N-H in plane bending vibrations. These assignments are in agreement with calculated and literature values (*Journal of Molecular Structure*, 1213 (2020) 128163; *Spectrochimica Acta Part A: Molecular and Biomolecular Spectroscopy* 129 (2014) 74–83).

Secondly, the simulated IR and Raman spectra of the Fmoc-F-F-His dimers 1-3 in Figure for review 1 are presented in Figure for review 2 and 3 together with the observed IR and Raman spectra (Fmoc-F-F (His)_Exp). Figure for review 2 indicates

that the simulated Raman spectra of the Fmoc-F-F-His dimers 1-3 are similar to the Raman spectra of Fmoc-F-F (His)_Exp since the calculated Raman vibrations of the Fmoc-F-F-His dimers 1-3 and the observed Raman vibrations of Fmoc-F-F (His)_Exp are mainly distributed in the range of 1000 - 1700 cm^{-1} and 2800 - 3200 cm^{-1} . Raman vibrations in the range of 1000 - 1700 cm^{-1} are mainly attributed to C=C and C-C stretching vibrational modes of the benzene ring, while the Raman vibrations in the range of 2800 - 3200 cm^{-1} are mainly attributed to C-H₂ stretching and C-H stretching modes of the benzene ring. Nevertheless, there are great differences in the range of 1000 - 1700 cm^{-1} between the simulated Raman spectra of the Fmoc-F-F-His dimers and the observed Raman spectra of Fmoc-F-F (His) Exp, and there are more vibrational modes in observed Raman curve because the corresponding structure of Fmoc-F-F (His) is much more complex than the three Fmoc-F-F-His dimers in Figure for review 1. The simulated IR spectra of the Fmoc-F-F-His dimers and the observed FTIR spectrum of Fmoc-F-F (His) are presented in Figure for review 3 and all the vibrational modes of both the simulated IR and the observed FTIR spectra are mainly distributed in the range of 1000 - 1700 cm^{-1} and 2800 - 3600 cm^{-1} . vibrations in the range of 1000 - 1700 cm^{-1} are mainly attributed to C=C and C-C stretching vibrational modes of the benzene ring and imidazole ring coupling with C=N and C-N stretching, while the vibrations in the range of 2800 - 3600 cm^{-1} are mainly attributed to C-H, N-H and O-H stretching modes. (*Spectrochimica Acta Part A: Molecular and Biomolecular Spectroscopy* 129 (2014) 74 - 83). Similar to the analysis of Raman spectra, there are great differences between the simulated IR spectra of the Fmoc-F-F-His dimers and the observed FTIR spectrum of Fmoc-F-F (His), and there are more vibrational modes in observed FTIR curve because the corresponding structure of Fmoc-F-F (His) is much more complex than the three Fmoc-F-F-His dimers.

Figure for review 2 The simulated Raman spectra of the Fmoc-F-F-His dimers 1-3 and the observed Raman spectrum of Fmoc-F-F (His).

Figure for review 3 The simulated IR spectra of the Fmoc-F-F-His dimers 1-3 and the observed FTIR spectrum of Fmoc-F-F (His) co-assembly.

2. Add some references supporting the Raman and Infrared spectral analysis results.

Journal of Molecular Structure, 1213 (2020) 128163; Journal of Molecular

Liquids, 290 (2019) 111209; Spectrochimica Acta Part A, 105 (2013) 218-222.

Response: Thanks for the excellent suggestion. Accordingly, we have made the following the revisions in the last paragraph of page 6-7: Second, based on Raman spectroscopy (**Figure for review 1**), the strong peak of Fmoc-F-F (His) at 1270 cm^{-1} illustrated violent vibration of the benzene ring or asymmetric stretching of carboxylic acid under the influence of His. Consistently, Fourier-transform infrared spectroscopy (FTIR) identified that the benzene band at 1450 cm^{-1} , representing the C=C vibration of benzene ring, (*Journal of Molecular Structure 1213 (2020) 128163*) significantly decreased in the presence of His. In addition, Fmoc-F-F exhibited stronger C-N stretching (1030 cm^{-1}) and N-H (739 cm^{-1} , 3300 cm^{-1}) (*Spectrochimica Acta Part A: Molecular and Biomolecular Spectroscopy 129 (2014) 74–83*) bending compared with Fmoc-F-F (His) and His. Weakening of the Fmoc-F-F (His) amines indicated the formation of non-covalent interactions. The red shift of C=O (1630 cm^{-1}) implicated the involvement of -COOH in hydrogen bonding.

3. Add some references related to the title compound in the introduction part.

Chem. Rev. 2021, 121, 22, 13869–13914; Spectroscopy letters 47 642–648 (2014).

Response: Thanks for the reviewer's recommendation. These references are very useful for us to learn the progress in the field of peptide assembly and His catalysis. We have added the references to the first paragraph and the third paragraph in introduction section, respectively.

Reviewer #2

The manuscript by Ye Yuan et al., describes the characterisation of the FMOC-FF with and without the addition of Histidine amino acid. They also explore the effect of other amino acids on morphology. The authors then characterise the ability of the mixture to catalyse a reaction. Finally, the author examine the catalytic ability of Amyloid-beta 1-40 and 1-42. Overall this is a thorough and extensive investigation of the phenomenon of mixing FMOC-FF with His and the outcome of this. The authors suggest that this is a co-assembly although it remains uncertain whether they are able to distinguish between

self-sorting and co-assembly. What is clear though is that the HIS has an influence on morphology of the FMOC-FF and that the presence of HIS confers some ability to generate ROS. It appears that His alone is quite active in generating Oxygen. The authors then move on to the separate and unrelated peptide, Amyloid-beta (1-40 and 1-42) to investigate whether it is also able to catalyse the reaction. The generation of ROS by Amyloid-beta has been shown before. The experimental work is well done. The catalytic assays are suggestive of catalysis but the thorough enzyme kinetics analysis is missing. The paper includes very nice CryoEM images but analysis of the data is missing.

I have a number of comments that must be corrected and addressed

Response: Thanks for the critical suggestion. The generation of ROS by Amyloid-beta has been shown before, but most of work proposed that the catalysis is from the chelated metal ions, such as iron, zinc, copper, etc, while the A β aggregate itself is inert. In contrast, in our work we first discovered that ROS could be directly generated by A β through the intrinsic enzyme-like catalysis. All suggestions have been carefully addressed revised.

1) FMOC-FF is not anything like Amyloid-beta. It does not mimic Abeta and shares very little sequence similarity (two amino acids). Amyloid beta does not possess the large FMOC and I do not believe that the authors can claim similarity between a small molecule dipeptide with FMOC and a 42 amino acid protein. Therefore, this needs editing throughout. There is no evidence that FMOC-FF is able to self-assemble in any similar way to the large peptide. There is no evidence that FMOC-FF is able to give any information about the behaviour of Abeta 1-42 or to Alzheimer's disease mechanisms. therefore, the paper should be edited. The FMOC-FF section should separate from the Abeta section. e.g. The first sentence of abstract is incorrect. How is a FMOC-FF small molecule "derived" from Amyloid beta. Many proteins have FF present. We can not claim that FF "derives" from these either. "Alzheimer's disease (AD) amyloid-beta (A β) polypeptide-derived diphenylalanine peptide (FF)" - remove.

Also from the title, which must be corrected "dipeptide assembly of A β motif" "and may not fully mimic the A β assemblies that occurs in the physiological environment". I would suggest it doesn't mimic it at all and this should be removed.

Response: Thanks for the constructive suggestion. A β aggregates in AD provide a model to develop peptide assembled nanostructure and functional nanomaterials. Herein, we referred to the previous studies of Ehud Gazit, who first reported on phenylalanine dipeptide as the core recognition motif of Alzheimer's β -amyloid and the smallest assembly unit of amyloid protein aggregation (*Science* 2003, 300, 625-627). Since then, many studies have described that F-F motif could be selected as an ultrashort amyloid model (*Chemical Communication* 2006, 2332-2334; *ACS Nano* 2020, 14, 6, 7181–7190). The precise mechanism of A β amyloid formation at the molecular level remains poorly understood due to the complex molecular structures of biological proteins and peptides and in some cases a mixture of secondary structural conformations. Such kind of simplified model may help to study the underlying molecular details of conformational changes and potentially setting the basis for rational molecular design of inhibitors or modulators to interfere amyloid aggregation process (*ACS Nano* 2017, 11, 6, 5960–5969). Fmoc-F-F can self-assemble to form highly ordered and stable β -sheet-rich fibrous nanostructures via intermolecular hydrogen bonding and aromatic stacking interactions, showing the characteristic morphology, secondary structure conformation, and Congo red/ThT binding signatures, which is similar to assembly features of amyloid fibrils.

Due to its chemical simplicity and versatility, F-F rapidly became the paradigm for the study for peptide self-assembly in functional nanomaterials which show broad potential in biomedical applications. In this paper, our original purpose is to develop peptide assembled nanozymes to conduct catalysis for substrates of natural enzymes. Distinct from previous Fmoc-F-F assembly in chemical hydrophobic condition, we tried to use amino acids to regulate the assembly under aqueous condition, among which His showed the effect to modulate Fmoc-F-F assembly into nanofilament. His is also a critical residue in the active center of natural enzymes, especially in

peroxidase (*Chemical Communications* 2017, 53 (2): 424-427). We fully agree the reviewer's point that Fmoc-F-F cannot follow the same assembly rules of A β due to the huge differences of sequence and reaction condition. However, the catalytic activity of Fmoc-F-F/His assembled nanofilament as a nanozyme provides us an inspiration that A β assembly may also have similar catalytic activity and act as a nanozyme in biological system. In particular, besides, F-F (19 and 20), A β also contains three His (6, 13, 14). In addition, the hydrophobic LV (17 and 18) in LVFF (A β 17-20) in A β may serve similar function of Fmoc and facilitate similar assembly into nanofilament as hydrophobic Fmoc group in Fmoc-F-F. These common features make it possible to form similar catalytic site in A β aggregates. In previous studies, A β aggregate in AD provides inspirations of fundamental principle and molecular basis for Fmoc-F-F assembly nanostructures. In our work, the catalytic feature of Fmoc-F-F assembly (with His) as nanozymes in turn inspires the investigation on catalytic function of A β deposit to help understand AD pathogenesis.

2) Fibrosis is not the correct word to use. Fibrillogenesis is preferable. Fibrosis means something entirely different. This is found both in the main text and also in the Suppl. Throughout the text requires careful proof reading including in the abstract. e.g longer time aggregation etc.

Response: Thanks for pointing out this problem. We have made careful proof reading and corrected them in the revised manuscript.

3) I think the authors mean sonication, rather than "ultra-sonic process"?

Response: Thanks for pointing this mistake. We have corrected them in the revised manuscript.

4) Can a three component (FMOC and two amino acids) really be claimed to form a-helical or beta sheet structures? Given the expected 3.6 residues per turn for an alpha helix? I don't think this is reasonable to claim this small molecule forms protein-like conformations.

Response: We thank the reviewer's constructive question. "Given the expected 3.6

residues per turn for an alpha helix” were studied in textbooks. In recent years, owing to the study of self-assembly of short peptides, the generalized secondary structure has also been found and defined. As shown in the reference of “Charge-Induced Secondary Structure Transformation of Amyloid-Derived Dipeptide Assemblies from β -Sheet to α -Helix” (*Angewandte Chemie* 2018, 57, 1537-1542), the amyloid-derived dipeptide Fmoc-FF can form different secondary structures of β -sheet and α -helix under different pH conditions.

5)"involuntarily" should be replaced with an alternative. This implies intention and doesn't make sense in this context.

Response: Thanks for pointing out this issue. The inappropriate word has been deleted in the revised version of the manuscript.

6) When the authors say Cystine, they mean cysteine (the amino acid) not the disulphide bonded form which is cystine.

Response: Thanks for pointing out this mistake. We have corrected it as Cysteine in the revised manuscript.

7) the authors showed cryo EM showing a regular fibrillar structure - could they have done more analysis on these structures to gain atomic information on the arrangements. What is the pitch? What is the significance of the measurements? these have not been put into context of the structural organisation of the small molecule.

Response: Thanks for the constructive suggestions and questions. We have made the efforts to obtain the three-dimensional structure of the co-assembly, but failed to conduct reconstruction analysis. The reason may be that the fibrillar structure is tangled too tightly to gain atomic information on the arrangements. We tried to reconstruct a possible three-dimensional structure, but TOMO results verified that it was wrong. In fact, it is still a big challenge to resolve 3D structure of peptide-assembled nanofibers and most of their structures are predicted based on raw morphologies rather than precise experimental measurements. For our work, based on

the cryo-EM, we speculated that the whole nanofilament is assembled by forming connective spindle units which consist of main structure with parallel bundles and entangled bundles at the ends. As shown in Fig. 2b, the bundle winding pitch between the wires is about 132 Å. Overall, cryo-TEM further confirmed that His changed the aggregation of Fmoc-F-F and formed co-existing tangled nanofilaments from stacked nanorods.

8) Page 7 we are told "Third, Fmoc-F-F (His) crystal peak diffraction intensity was high, sharp, and scattered, indicating a good crystal state. In addition, the fewer peaks of Fmoc-F-F (His) accompanied by a shift in the diffraction peaks suggested that His may enter the original crystal lattice and cause distortion. Strong diffraction peaks at 21° and 24° were formed (Supplementary Fig. 11)." But there is no intro to this technique - is this single crystal X-ray crystallography? No Data is shown. Is this fibre diffraction? More information is required here to make sense of this information.

Response: Thanks for your suggestion. These analyses were conducted using X-ray diffraction (XRD) to investigate the interaction between Fmoc-F-F and His. XRD is a versatile method used commonly in the field of nanotechnology to characterize nanomaterials regarding the composition, crystal structure, and crystalline grain size. We have added the XRD information in the revised manuscript and supplementary information.

9) p11. what is DFT?

Response: Thanks for the opportunity to clarify this issue. Density-functional theory (DFT) is a computational quantum mechanical modelling method used in physics, chemistry and materials science to investigate the electronic structure, in particular atoms, molecules, and the condensed phases. Its goal is the quantitative understanding of material properties from the fundamental laws of quantum mechanics. DFT simulation can calculate a vast range of structural, chemical, optical, spectroscopic, elastic, vibrational and thermodynamic properties, and it is nowadays common

practice to include computational results in experimental studies on materials and surfaces.

Based on our experimental characterizations, DFT calculations were performed to analyze the intermolecular interactions and molecular organizations between Fmoc-F-F and His molecules, including the π - π stacking of Fmoc-F-F dimers and Fmoc-F-F-His dimers connected by different hydrogen-bonding interaction modes with corresponding binding energy as listed in Fig. 4.

Besides, DFT calculations were also used to understand the plausible mechanism of POD-like catalysis of Fmoc-F-F (His) as presented in Fig. 6 based on the three reactions (eq 1, eq 2, and eq 3). Our calculations indicated that the H_2O_2 molecule binds to the imidazole rings via hydrogen bonding with a binding energy of 0.63 eV in the catalytic center, as shown in state (2) in Fig. 6. The hydrogen bonding interactions between H_2O_2^* and imidazole rings facilitate the rupture of the O-O bond in H_2O_2^* . The adsorbed H_2O_2 molecule in the catalytic center firstly breaks the O-O bond, and one of the dissociated OH radicals bind to the H atom from TMB to form a H_2O^* molecule and oxTMB* molecule. After the escape of H_2O^* and oxTMB*, the remaining OH* radical in the catalytic center oxidizes the next TMB molecule. As shown in Supplementary Fig. 33, under acidic conditions, the first TMB molecule is easily oxidized by H_2O^* with an energy barrier of 1.41 eV in step 2, while the second TMB molecule is easily oxidized by the OH* radical, as shown in step 3. From DFT analysis, the POD-like catalytic center of the co-assembly of Fmoc-F-F (His) effectively catalyzes the oxidation of TMB by H_2O_2 under acidic conditions due to the arrangement of His molecules in Fmoc-F-F (His) facilitating POD-like activity.

In this manuscript, although DFT calculations may not fully explain the experimental phenomena of the co-assembly effect on Fmoc-F-F by the addition of His, it helps us better understand the complex molecular mechanism of co-assembly and provide insights into the POD-like catalytic mechanism of Fmoc-F-F (His) and A β filament.

10) Enzyme kinetics is not shown and I think this is needed to clearly show

catalysis

Response: Thanks for pointing out this problem. Enzyme kinetics of Fmoc-F-F (His) and A β filament were tested and have been updated in the revised manuscript. Both of them followed the typical enzymatic characteristics (Figure for review 4 and 5). As for Fmoc-F-F (His), the K_M (the Michaelis constant, representing the affinity of the enzyme for the substrate) of Fmoc-F-F (His) were calculated to be 5.61 mM for H₂O₂ and 0.919 mM for TMB. As for A β filament, the K_M (the Michaelis constant, representing the affinity of the enzyme for the substrate) of Fmoc-F-F (His) were calculated to be 0.40 mM for H₂O₂ and 0.32 mM for TMB. A β filament had a good affinity for H₂O₂, which indicates that it may exert oxidative damage *in vivo*.

Figure for review 4 Kinetics assay of Fmoc-F-F (His). **a**, Michaelis-Menten curves against H₂O₂ concentration, in which the concentration of TMB was fixed at 1.0 mM; **b**, Michaelis-Menten curves against TMB concentration, in which the concentration of H₂O₂ was fixed at 10 mM. (Please also see **Supplementary Fig. 24** in the revised supplementary information)

Figure for review 5 Kinetics assay of A β filament. **a**, Michaelis-Menten curves against H₂O₂ concentration, in which the concentration of TMB was fixed at 1.0 mM; **b**, Michaelis-Menten curves against TMB concentration, in which the concentration

of H₂O₂ was fixed at 10 mM. (Please also see **Supplementary Fig. 29** in the revised supplementary information)

11) I could not find any information about the relative amounts of FMOC-FF and His? What is the ratio?

Response: Thanks for pointing out these problems. The feeding mass ratio of Fmoc-F-F and His were 1:10 (molar ratio: 1:34.5). Different mass ratios of Fmoc-F-F and His (1:1, 1:5, 1:6, 1:7, 1:9, 1:15) in which the concentration of Fmoc-F-F was fixed at 2 mg mL⁻¹ were investigated in Supplementary Fig. 4, and different mass ratios of His and Fmoc-F-F (40:1, 20:1, 6.67:1, 2:1) in which the concentration of His was fixed at 20 mg mL⁻¹ were investigated in Supplementary Fig. 5. Through comparison and screening, we selected the most appropriate proportion which could form uniform filament (The feeding mass ratio of Fmoc-F-F and His were 1:10).

12) Figure showing Alpha-fold prediction needs further explanation. If the structures with His6Ala is different, it could be the reason for different activities (rather than missing His). Plus there are two additional HIS residues within Abeta. Why are these not considered - especially since they are consecutive, His-His?

Response: Thanks for your constructive suggestions. We agree the point that activity reduction may be caused by structure change and then analyzed the structure changes after His mutant with alphafold2. Besides F-F y motif, A β also consists 3 His (6, 13, 14). We firstly customized peptides synthesis with different amino acid mutations from the company before we submitted the manuscript, but only A β 1-42 (6His→Ala) peptides could be obtained. When the mutant sequences were proposed, the company arranged to synthesize the mutant immediately but mass spectroscopy identified that the products were impure and not as-expected mutants. We have tried them with the GL Biochem (Shanghai, China), Genscript Biochem (Shanghai, China), Sangon Biotech (Shanghai, China) and been informed that β amyloid peptide with mutation is difficult to synthesize, especially 13 or 14 His (close to the drainage area).

In order to determine the relationship between A β 1-42 and His, different structures of mutants were predicted by alphafold2 (Figure for review 6). For the convenience of comparison, we choose tetramer as the basic assembly model. Consistent with reported crystal structure, the predicted A β 1-42 could form symmetrical β -sheet structure. 6His was existed in the random coil area. The intermolecular distance between 6His and 19Phe-20Phe is 10.28 Å in tetramer model, which is much shorter than that of 13His and 14His (15.58 Å), indicating that 6His may be more important for the construction of catalytic center of peroxidase-like activity. If all His were mutated (6, 13, 14His→Ala), A β 1-42 could not form any stable secondary structure except random coils, indicating that His is important for forming β conformations and may regulate A β 1-42 assembly. The A β 1-42 with 6His→Ala mutant is composed of two large reversals β pieces and two small β -structural composition, which is similar to the structure of wild-type A β 1-42, indicating such mutation may not dramatically change the structure.

In contrast, the A β 1-42 with 13His→Ala mutant is composed of two large reversals β pieces and one small β -turn (in the position of 6His), and the distances of 6His and 14His to 19Phe-20Phe were 10.55 Å and 16.12 Å, respectively. The A β 1-42 with 14His→Ala mutant also contains two larger reversals β pieces, in which the distances of 6His and 14His to 19Phe-20Phe were 7.55 Å and 14.52 Å, respectively. Overall, based on the predicted structure, the mutation of His in A β would affect the structure and assembly on different degree. In particular, 6His is close to F-F in assembled structure and critical to form catalytic site. In contrast, 13His and 14His seem to have few contribution on catalysis and their mutants may not affect the interaction between 6His and F-F. For future study, we will continue to try the preparation of His-mutant A β peptides and verify our hypothesis with experiments.

Figure for reviewer 6. AlphaFold2 predicted assembled tetramer structure of Aβ₁₋₄₂ and different His mutants. The distance between His and F-F (19-20) are marked. (Please also see **Supplementary Fig. 30** in the revised supplementary information)

12) Supplementary section requires careful proof reading throughout

Response: Thanks for the constructive suggestion. We have checked Supplementary section and carefully revised it.

Reviewer #3

In this study, the authors used Fmoc-F-F peptides with/without His to study their propensity to form fibrillar aggregates. They found that His has a property to modulate the physical properties of Fmoc-F-F dipeptides. Next, inspired by this findings, they studied the possible POD-like activity of Ab1-40 and Ab1-42 because the F-F motif is seen in the amyloid fibrils made from these proteins. They successfully discovered that Ab1-40 and Ab1-42 fibrils show POD-like activity, thereby producing ROS species. They discuss that the enzymatic activity of Ab fibrils may be related to the pathogenesis of AD. The findings presented in this paper will shed new light on the molecular mechanism of the pathogenesis of AD. To the reviewer's knowledge, this is the first study to discover that amyloid fibrils show enzymatic activity. This work will play an important role in elucidating the mechanism of amyloidosis. On other other hand, there are some points that the authors should focus on. I will describe these points below. In particular, analysis and interpretation of small-angle X-ray scattering data should be done correctly.

Response: Thanks for careful review and positive comments from the reviewer. All suggestions have been seriously addressed.

Major:

1. Both Fmoc-F-F and Fmoc-F-F(His) increase ThT fluorescence intensity, suggesting that both forms amyloid-like structures. To better understand the mechanism of fibril formations, the authors should study the time-course of ThT intensity changes during fibril formations for both samples.

Response: Thanks for the constructive suggestions. The time-course of ThT intensity changes during fibril formations for both samples were tested (Figure for review 7). We found that the fluorescence intensity showed an upward trend with the continuous progress of the co-assembly. The experimental results confirmed that His could regulate the formation of fibril, which was accompanied by the continuous

intensification of co-assembly. We have updated this information in the revised manuscript.

Figure for review 7 ThT-binding assay proved the formation of amyloid-like filaments. (Please also see **Supplementary Fig. 15** in the revised supplementary information)

2. Relating to this, to obtain more consistent results with measurements using Ab, POD-activity of intermediate oligomer states of Fmoc-F-F and Fmoc-F-F(His) should be investigated. This will clarify whether the POD-activity of the minimal motif is indeed closely related to the POD-activity of Ab proteins because the authors already detected that Ab fibril POD-activity is stronger than Ab monomers and oligomers.

Response: Thanks for the constructive suggestions. POD-activity of intermediate oligomer states from Fmoc-F-F to Fmoc-F-F (His) were investigated We found that the POD-activity showed an upward trend with the continuous oligomer states of the co-assembly (Figure for review 8). We have updated this information in the revised manuscript.

Figure for review 8 POD-activity of intermediate oligomer states of Fmoc-F-F (His). (Please also see **Supplementary Fig. 23** in the revised supplementary information)

3. 208-211 Treatment of the SAXS data is wrong. The value of -0.03758 means nothing. Cross-sectional Guinier plots or Guinier plots should be applied to this data first. Then, the cross-sectional radius of gyration (R_c) or the radius of gyration (R_g) values should be estimated. Further, the equation used for (cross-sectional) Guinier plots should be described. Supplementary Fig. 19 The SAXS curve should be displayed in the logarithmic scale. In the inset, the scale of the ordinate should be changed such that the curve is more clearly seen. Both the measured data points and the fit should be displayed. The abscissa should start from $q^2=0.0$ [nm⁻²]

Response: Thanks for pointing out this critical problem. We re-evaluated and revised the analysis of SAXS data. Guinier plots of the scattering data are presented in Figure for review 9. Besides, the radius of gyration (R_g) value, estimated from the linear parts using Guinier analysis, is 11.89 ± 0.17 .

Figure for review 9 The fitting of the scattering data using Guinier analysis. (Please also see **Supplementary Fig. 17** in the revised supplementary information)

In the Guinier region, the SAXS intensity can be approximated by:

$$I(Q) = I(0)\exp\left(-\frac{Q^2 R_g^2}{3}\right) \quad (1)$$

Equation (1) is suitable for monodisperse systems. A plot of $\ln[I(Q)]$ versus Q^2 reveals the value of R_g from the slope (Glatter and Kratky, 1982):

$$\ln[I(Q)] = \ln[I(0)] - \frac{R_g^2}{3} Q^2 \quad (2)$$

Herein, R_g is defined as the root-mean square of the distances of all electrons from the center of mass. R_g can be obtained from SAXS measurements, using the Guinier approximation.

The SAXS curves are displayed on a logarithmic scale and the scale of the ordinate/abscissa were adjusted in Figure for review 10.

Figure for review 10 The scattering curves are displayed in logarithmic-intensity scale. (Please also see **Supplementary Fig. 18** the revised supplementary information)

All above data have been updated in the revised manuscript and supplementary information.

4. 287-290 The authors found that Ab fibrils show higher POD-like activity compared with monomers and oligomers. Does this mean that fibrils are more cytotoxic than oligomers? The authors should discuss this point.

Response: Thanks for the constructive suggestions. A β aggregation is the initiating mechanistic event, in which the different stages of aggregates, from soluble oligomers to insoluble fibrils in plaques, are believed to impair synaptic function and ultimately damage neurons, resulting in chronic neurodegeneration and cognitive impairment and finally dementia (*Trends Pharmacol Sci.*, 36, 297-309(2015)). The cytotoxicity of A β 1-42 monomers, oligomers and filaments were tested using HT-22 and the results showed that their cytotoxicity followed the order filaments> oligomers> monomers (Figure for review 11). The A β 1-42 filaments performed the higher POD-like activity, which may increase ROS level to damage neuron cells and induce inflammatory response. In particular, such A β 1-42 filaments as nanozymes are stable and can persistently perform such catalytic activity, serving as a long-acting catalytic unit. We thus hypothesized that the POD-like activity may provide a link between A β 1-42 assembly and its cytotoxicity.

We also noticed that the current amyloid cascade hypothesis in AD has transferred from A β plaques to soluble A β oligomers which can exert the toxicity through receptor and direct membrane interactions (*Nature Neuroscience* 2012, 15: 349–357; *EBioMedicine*. 2016, 6: 42–49). Such change may be due to lacking direct correlation between A β fibrils and the loss of synapses and neurons in brains. Our findings indicate that A β fibrils may directly contribute to AD pathogenesis by acting as nanozymes to exert persistent catalysis. For future study, we will test the A β plaques from AD patients to validate their catalysis.

Figure for review 11 The cytotoxicity of different states of A β 1-42 to HT-22 cells. (Please also see **Supplementary Fig. 31** in the revised supplementary information)

5. 420-421 Sample concentrations, quantities, exposure times, and how the data reduction was conducted should be written here for SAXS measurements.

Response: Thanks for pointing out these issues. Solid samples were used for testing. Exposure times were set as 3600 s. The manuscript has been revised.

Minor:

6. 34-37 This sentence can read as if Ab causes Parkinson's disease as well as AD.

Is it true?

Response: Thanks for carefully checking. The main pathogenesis of Parkinson's disease (PD) and AD were different, but both of them could be affected by A β . A β deposition was one of the main hypotheses of the pathogenesis of AD. the self-assembly of misfolded proteins and peptides into highly ordered β -sheet-rich oligomeric structures and further into fibrillar aggregations, which are known as amyloid fibrils, is associated with various degenerative disorders (*ACS nano*, 2020, 14, 7181–7190). PD is also a type of amyloidosis disease (*PNAS* 2019, 116 (36):

17963-17969) and A β deposition is positive comorbid association with cognitive decline in PD (*Journal of Neurology* 2019, 266: 2605–2619). It is reported that A β deposits dramatically accelerated α -syn pathogenesis and spread throughout the brain, showing an overlap of A β plaques, tau tangles, and α -synuclein (α -syn) pathologies in the brains of AD and PD (*Neuron* 2020, 105(2):260-275).

Reviewer #4

This is a thorough and interesting study, combining experiment and molecular that deserves publication. The extrapolation of the properties of the FF assembly structure to Abeta assembly and neurodegenerative diseases is a bit far fetch. This said I think the results in itself is an interesting example of nano-construct that can have a catalytic activity. It should be termed catalytic activity not enzymatic activity as it lacks the specificity and selectivity of a full-fledged enzyme. I think the term nano-enzyme is very misleading.

I do not think the results are general and urgent enough to be published in Nature Communication. I would suggest publication in a more physical chemistry oriented journal. For such a publication it would be interesting to investigate the influence of ionic strength and buffer composition on the formed nano-structures. Such information is crucial in order to assay the applicability of the formed nano-structures. It would also give further insight on the forces that regulates the assembly process.

Response: Thanks for the systematic evaluation and suggestions of the reviewer. We appreciate the positive comment for our work as a thorough and interesting study. The original purpose in our study is to develop amino acid/peptide based nanozymes to mimic activity of natural enzymes. Nanozymes are an emerging field connecting biology and nanotechnology, drawing a lot attentions ranging from fundamental research of developing nanoscale artificial enzymes to transitional research of developing new theranostic platforms for diseases. In recent years, many nanomaterials are found with catalytic activity that can catalyze the reactions of

biochemical substrates mediated by enzymes. To distinguish them from traditional nanocatalysts focusing on chemical catalysis in industry, nanozymes are used to define these nanomaterials with enzyme-like catalytic properties and promote their biomedical applications ranging from *in vitro* biosensors and immunoassays into *in vivo* tumor therapy, antibacterial, antioxidant (*Accounts of Materials Research* 2021, 2(7): 534–547; *Chemical Society Reviews* 2019, 48: 1004-1076). Currently, nanozymes are proposed as a new generation of artificial enzymes, although their catalytic mechanism may not be completely same as those of natural enzymes (*Nano Today* 2021, 40: 101269). Different to natural enzymes, nanozymes have multiple active sites on their surface and their activity can be determined not only by active site structure and number, but also by the nanoscale size, morphology and surface modification. In previous studies most nanozymes are made from inorganic metals, metal oxides, carbon/graphene or inorganic-inorganic hybrids (e.g. metal-organic frameworks).

To improve their catalytic activity and biocompatibility, recent studies including ours found that mimicking the structure of active center of natural enzymes or directly introduce key amino acid residues can significantly improve the activity of nanozymes (*Chemical Communications* 2018, 9 (1): 1440; *Advanced Functional Materials* 2021, 31 (9), 2007130; *Nature Catalysis* 2021, 4: 407–417). Following this strategy, we made a further exploration to develop nanozymes directly using amino acids and peptide by self-assembly. It has been known that His is a key residue in many active centers of natural enzymes and can be used for assembly to mimic enzymatic catalysis (*Nature Materials* 2021, 20(3):395-402). In particular, His is frequently present in the active center of peroxidases which catalyzes the decomposition of H_2O_2 into $\cdot\text{OH}$ radicals. In addition, F-F is a moiety with self-assembly ability owing to intermolecular hydrogen bonding and π - π stacking. The assembly is preferred to conduct under aqueous condition in order to improve biocompatibility. Regarding to such purpose, we investigated the interaction between amino acid (His is the major one) and Fomc-F-F in the assembly process of nanofilament and enzyme-like property as a nanozyme candidate.

Our work for the first time provides an amino acid/peptide assembled nanozyme, which not only extends the material type in the field of nanozymes, but also provides a new strategy for rational design of nanozymes. At the beginning, we are excited to find the rule of assembly and the peroxidase (POD)-like activity of such nanofilament nanozymes. Following in-depth investigation and learning more literatures, we noticed that using Fmoc-F-F to assemble nanostructure is inspired from A β assembly as the core dipeptide motif. Such literature further inspired us if A β assembly performs similar catalytic activity as the His/Fmoc-F-F nanofilament. We think that this investigation is very significant as it has been known that A β deposit has strong relationship with AD pathogenesis but the mechanism is yet fully appreciated. The finding of peroxidase-like property of A β filament further extends the study of nanozymes, which demonstrates that there are natural nanozymes in biological system and may play important role in many physiological or pathological processes. Importantly, it provides new insight for understanding the role of A β assembly in AD pathogenesis, that is, acting as a nanozyme to catalyze ROS formation and thus damage neuron. This nanozyme mechanism may help AD prevention by introducing antioxidant intervention.

Therefore, we think our findings not only extend the research in the field of nanozymes, but also help understanding the in-depth relationship between A β and AD. The two fields seems in a far fetch, but our work demonstrates that the interdisciplinary combination via fundamental research of nanozymes, peptide assembly can be achieved to promote each other. In fact, besides AD, many protein aggregates have been found in many diseases, such as islet amyloid polypeptide in Type II diabetes, human calcitonin in thyroid carcinoma, α -synuclein in PD, antioxidant enzyme SOD1 in ALS. Our work may provide a new view to study their role in pathogenesis. We think our findings will be significant and show broad readership in the fields of nanozymes, peptide assembly of nanostructure, artificial enzymes, AD, protein aggregates-related diseases.

Reviewer #5

In the biochemical and cell culture assays of the study, the authors investigated the effects of A β peptides (A β 1-40 and A β 1-42) on oxidative stress and reactive oxygen species (ROS) productions in the mouse hippocampal (HT-22) cell line and bilateral hippocampus of SD rats. The generation of ROS was increased in the cell line and rat hippocampus by increasing the POD activity, although the activity of catalase was decreased. The authors observed that A β 1-42 filaments induced ROS generation actions due to their POD activities. The study is potentially interesting for the journal, and it has also high citation potential. However, it is careless presentation. There seem to be areas of considerable scientific confusion in the sections of methods, results, and discussion, especially. Before making a judgment regarding the document, the author needs to respond appropriately to the main criticisms listed below.

Response: We really appreciate the reviewer's understanding of the merits of the present study and the positive comments.

Major remarks

1. In 'Cellular and in vivo ROS assessment'. The section is too short. For example, how many cells such as 1×10^6 and 2×10^6 cell were used for the analyses? Were the HT-22 cells obtained? For the dose of $10 \mu\text{l}$ A β filaments (1mg mL^{-1}), please add a reference or reason. For obtaining the hippocampal neurons, please add a reference or reason. There are no details of gender and age of animals in the section.

Response: Thanks for pointing out these problems. We have revised this section with more experimental details. For ROS fluorescence intensity detection, 10,000 cells were collected and analyzed by flow cytometry for each test. We used HT-22 (SCC129) which is an immortalized mouse hippocampal neuronal cell line for cellular ROS assays. For in vivo assay, A β -mediated oxidative stress damage is considered a direct cause of Alzheimer's disease, which usually occurs in the hippocampus

(*Neurobiology of Aging*, 2014, 35(3):472-81). The detailed procedure for collecting hippocampal neurons has been added in the revised manuscript with reference (*Neurological Research* 2019, 41: 77-86). The dose of 10 μ L A β filaments (1 mg mL⁻¹) was referred to the procedure reported by previous paper (*Neurol Sci.* 2014, 35, 35-40). The reason for choosing male SD rats is that estrogen may have neuroprotective and neuroenhancing functions (*Annals New York Academy of Sciences*, 2001, 949:223-34). Thus, in the construction of AD model, male animals are preferred. We have revised the related information and added the references in the revised manuscript.

2. Mitochondria have great importance on the generation of ROS in the cells and neurons. The detection of mitochondrial ROS by using a probe MitoSOX may be useful for the current study?

Response: Thanks for the constructive suggestion. Mitochondrial ROS was investigated after treatment of A β filament using HT-22 (Figure for review 12). Detailed methods have been supplemented in the revised manuscript. Similar results could be obtained contrast to cROS (Fig. 5g) and lipid ROS (Fig. 5h). We confirmed that A β 1-42 has ability to increase ROS levels in HT-22 cell.

Figure for review 12 ROS of mitochondria in HT-22 cells treated by different concentrations of A β 1-42 filaments. (Please also see **Supplementary Fig. 32** in the revised supplementary information)

3. Discussion section is too short and it lacks focus. The author should concentrate on interpretation of their findings and their relevance to the field of study. The discussion should be used for the interpretation of data and for pointing out the significance of the findings.

Response: Thanks for the constructive suggestion. We have revised the discussion section. We discussed that the assembly-driven catalytic feature of Fmoc-F-F (His) nanofilaments make them to be a new type of nanozymes which can catalyze biochemical reaction of the substrate of enzymes and thus perform enzyme-like activity as a new generation of artificial enzymes. Moreover, we highlighted that A β 1-42 filaments might also be a nanozyme due to their similar catalytic activity and core motif of peptide and amino acid mediated assembly. As A β 1-42 filaments are assembled by natural polypeptide and present in AD, we emphasize that A β 1-42 filaments are a type of natural nanozymes which are protein in nature but the catalytic activity is based on protein assembly to form multiple active sites in one filament. Although showing similar apparent enzymatic kinetics, such natural nanozymes are different from traditional enzymes which are also protein in nature but typically have single active center in one enzyme. We thus pointed that natural nanozymes may be ubiquitous catalytic supramolecules in biological system and helpful to understand the role of numerous protein aggregates in many diseases. For more detailed information, please see the revised manuscript.

4. There are serious problems in the abbreviations of the manuscript. The authors used both abbreviated and full names of several words such as ROS and Histidine through the manuscript.

Response: Thanks for pointing out these problems. The manuscript has been revised. We focused on distinguishing cROS, lipid ROS and ROS of Mitochondria. We

replaced His with Histidine.

5. DCFH-DA is a marker of cytosolic ROS. Please use ‘cytosolic ROS (cROS)’ instead of ‘ROS’ through the manuscript.

Response: Thanks for the suggestion. We checked the manuscript and revised them.

6. DCFH-DA is a fluorogenic probe that measures hydroxyl, peroxy and other ROS activity within the cell. The authors prepared hippocampus homogenate without collagenase and other procedures. The DCFH-DA is not suitable for the cell homogenate (injured cells), although it is suitable for HT-22 and whole hippocampal neurons. The authors should give detailed information on the preparation of hippocampal samples in the section.

Response: Thanks for pointing out the problem. In our experiment, the hippocampus cells were collected using trypsin digestion and their ROS levels were detected using flow cytometry. Meanwhile, the groups of sham (normal rats without treatment) and PBS (rats treated with PBS) were used to eliminate interference from the background and buffer. The detailed information on the preparation of hippocampal samples was provided.

1. After anesthesia, the rats were dislocated, sacrificed and the brain was collected (about 3-5 min);

2. After washing with normal saline, the right hippocampus was peeled off on the ice bag and quickly immersed in a dish containing PBS solution (2 min).

3. Then all the hippocampus samples were transferred into a 1.5 mL PE tube and incubated with trypsin (0.25%, 1 mL) for digestion in water bath at 37°C for 10 min;

4. The serum was added to stop digestion and the hippocampus cells were collected by filtration with 300 micron nylon mesh and centrifugation at 1000 rpm for 5 min;

5. Then the cells were resuspended with PBS and stained with DCFH-DA following the introduction of ROS detection kit. The cytosolic ROS was detected using a flow cytometry. Only cells in normal size was selected for analysis during flow loop gate.

We think that the above protocol would not cause significant damage to the cells and

thus the DCFH-DA probe is suitable for ROS detection. Similar method has been used by other investigation (*Neurological Research 2019, 41: 77-86*). We have added the above protocols in the revised manuscript.

In abstract

- In ‘...correlation with Histidine’. Please use ‘His’ instead of ‘Histidine’.
- Please delete once used ROS abbreviation in the ‘generate reactive oxygen species (ROS)’.

Response: Thanks for pointing out these issues. The manuscript has been revised.

In results

- In ‘SD Rats’, ‘Co-assembled Fmoc-F-F (His) nanofilaments exert enzyme-like activity to regulate ROS’. Please use the full name of the SD abbreviation before using it through the manuscript.
- In page 10, ... high POD-like activity which catalyzes TMB colorimetric reaction (the oxidized TMB with absorbance at 652 nm)... Please use 3,3',5,5'-tetramethylbenzidine (TMB) instead of ‘TMB’.
- In page 11,the oxidation reactions of the two 3,3',5,5'-tetramethylbenzidine (TMB) molecules.. Please use ‘TMB’ instead of ‘two 3,3',5,5'-tetramethylbenzidine (TMB).
- Please add p values to the section.

Response: Thanks for pointing out these issues. The manuscript has been revised correspondingly.

In material and methods

- ‘In ‘Cellular and in vivo ROS assessment’. Please use ‘ROS detection kit’ instead of ‘reactive oxygen species detection kit’.

Response: Thanks for pointing out this issue. The manuscript has been revised correspondingly.

In figure legends

- In Fig. 5. The unit of ROS and lipid ROS are different from the reference values. For ROS fluorescence intensity unit, arbitrary unit may be useful.

Response: Thanks for pointing the problem. The manuscript has been revised.

REVIEWER COMMENTS

Reviewer #2 (Remarks to the Author):

Several of my changes have been considered and updated in the manuscript and this is an improvement. I appreciate that cryoEM was not possible/

Im afraid I disagree that the authors have not updated and edited the manuscript regarding the similarity of the 40/42 residue peptide Amyloid beta and the FMOC-FF. I fundamentally disagree that this can be referred to as a "motif". The FF work referred to by the authors from Gazit is an FF, not FMOC-FF. Furthermore, I disagree that this is a motif of amyloid beta and it does not serve as a model system for the amyloidogenic, Alzheimer's peptide.

I do not agree that this paper should be published until this fundamental and important aspect has been addressed fully. It is misleading and unhelpful to report these two things as similar or part of the same unit.

I suggest that my original comments in point 1 should fully be addressed before publication

Reviewer #3 (Remarks to the Author):

The authors tried to improve the manuscript according to the reviewers' comments. I am afraid that regarding the SAXS part, the presented analysis does not report anything useful. The method of analysis and a way of presenting data are far from sufficient. Furthermore, what is the most critical is that the claimed R_g value is too small compared with the cryo-TEM structures shown in Fig.2. I wonder how these large objects can have the R_g of ~ 12 nm? This indicates that there is a severe problem in sample quality used for SAXS.

Reviewer #5 (Remarks to the Author):

The authors were performed all required comments on the cellular analyses and material in the MS, and I have no further comment.

The point-to-point response has been prepared as the followings:

Reviewer #2

Several of my changes have been considered and updated in the manuscript and this is an improvement. I appreciate that cryoEM was not possible.

I'm afraid I disagree that the authors have not updated and edited the manuscript regarding the similarity of the 40/42 residue peptide Amyloid beta and the FMOC-FF. I fundamentally disagree that this can be referred to as a "motif". The FF work referred to by the authors from Gazit is an FF, not FMOC-FF. Furthermore, I disagree that this is a motif of amyloid beta and it does not serve as a model system for the amyloidogenic, Alzheimer's peptide.

I do not agree that this paper should be published until this fundamental and important aspect has been addressed fully. It is misleading and unhelpful to report these two things as similar or part of the same unit.

I suggest that my original comments in point 1 should fully be addressed before publication

Response: We appreciate the reviewer's points on the relationship between Fmoc-F-F and A β . We agreed with the reviewer that FMOC-F-F is not suitable to be termed as the motif of A β peptide. We have removed the term "motif" in the title and the whole manuscript.

After carefully discussion with all the authors and learning from the literatures, we edited the logic of the manuscript by emphasizing that histidine modulates amyloid-like assembly (please see the introduction section in the revised manuscript). The amyloid-like aggregation have been found in many diseases and utilized to fabricate artificial nanomaterials (*Nat Rev Mol Cell Bio* 2014, 15, 384-396; *Chem Soc Rev*. 2017, 46(15): 4661-4708; *Front Bioeng Biotech*, 2021, 9, 641372). In the revised manuscript, FMOC-F-F is referred a modified dipeptide model that can assembly with amyloid-like feature of β -sheet structure to form nanofibrils in the presence of histidine. The interaction between histidine and FMOC-F-F not only rearranges the assembly behavior of FMOC-F-F, but also contributes to the peroxidase-like activity in the assembled nanofilaments. These results tell us that the assembly-like assembly may lead to unexpected biological properties for a peptide or protein under aggregate

state.

Such unique role of histidine in amyloid-like assembly inspired us to extend on the catalytic property of A β aggregates as a classical amyloid assembly. The A β 1-42 nanofilaments also demonstrated peroxidase-like activity. The structure analysis using Alphafold2 indicated that histidine (position 6) is close to F-F in the assembled A β 1-42. Such spatial proximity is intermolecular interaction in the assembled state. This mechanism indicates the impact of amyloid assembly in modulating the function of A β peptide, and also can be used to explain why free A β 1-42 monomer does not have peroxidase-like activity. Although such part is not fully investigated in our manuscript, the finding is meaningful to rethink the biological function of A β 1-42 aggregates in AD pathogenesis.

For future study, we will continue to study the relationship between assembly structure and enzyme-like property in A β aggregates and other assembled proteins reported in the related diseases. We hope that our work will provide fundamental knowledge on protein assembly in human diseases and for the fabrication of peptide nanomaterials with tailored properties, such as nanozymes with enzyme-like activities.

Reviewer #3

The authors tried to improve the manuscript according to the reviewers' comments. I am afraid that regarding the SAXS part, the presented analysis does not report anything useful. The method of analysis and a way of presenting data are far from sufficient. Furthermore, what is the most critical is that the claimed R_g value is too small compared with the Cryo-EM structures shown in Fig.2. I wonder how these large objects can have the R_g of ~ 12 nm? This indicates that there is a severe problem in sample quality used for SAXS.

Response: Thanks for your careful checking and comments. The previous calculation of R_g was based on Guinier plot fitted on the whole q range in the SAXS curve, which led to the too small R_g value. After learning the literatures and discussion with several experts in SAXS analysis, we have corrected the analysis for the Guinier approximation based on low q (see red line in **Figure for review 1** below). Then we obtained the value of R_g at 64.2 ± 2.4 nm. Such R_g value corresponds to the cylinder

scatters with the dimensions of diameter $2R \sim 15$ nm and length $L \sim 220$ nm (Based the correlation of $R_g^2 = (R^2/2) + (L^2/12)$). These results are well consistent with the Cryo-EM observation in Fig. 2. The detailed calculation information has been updated in the Methods section in the revised manuscript.

Figure for review 1 The SAXS data of Fmoc-F-F (His) nanofilaments. The scattering curve of fitted guinier plot was obtained at low q and represented by the red line. The R_g value was calculated based on the slope of Guinier plot. (Please also see **Supplementary Fig. 17** in the revised supplementary information)

Reviewers' comments:

Reviewer #2 (Remarks to the Author):

The authors have now made considerable efforts to amend the manuscript as suggested. This is now a well prepared manuscript that nicely demonstrates the capacity of FF filaments to assemble in the presence of HIS and also for Abeta fibrils to produce ROS.

Reviewer #3 (Remarks to the Author):

The authors claim that they reanalyzed the SAXS data using Guinier approximation. However, the way in which they carried out the analysis is unacceptable and they do not respond to the reviewer's comments appropriately. It is strongly advisable that how SAXS data are displayed and analyzed for elongated scattering particles. From the R_g value, they claim that R and L are consistent with other data, but the two parameters cannot be determined from one R_g value because it is an ill-posed problem. I already gave a clue in my first round of comments, but they ignored. In the current format, the SAXS results should not be published.

Reviewer #3

The authors claim that they reanalyzed the SAXS data using Guinier approximation. However, the way in which they carried out the analysis is unacceptable and they do not respond to the reviewer's comments appropriately. It is strongly advisable that how SAXS data are displayed and analyzed for elongated scattering particles. From the R_g value, they claim that R and L are consistent with other data, but the two parameters cannot be determined from one R_g value because it is an ill-posed problem. I already gave a clue in my first round of comments, but they ignored. In the current format, the SAXS results should not be published.

Response:

Thanks for the critical suggestions of the reviewer. Our objective in employing SAXS in our study is to further characterize the size dimensions of Fmoc-F-F(His) nanofilaments, thus providing complementary confirmation to the findings obtained through Cryo-EM and TEM. We are sorry for not interpreting the SAXS data properly in the previous response, as we combined the computer fitting and Guinier analysis in a confusing way. We have carefully considered the clues you provided and made changes to the manuscript as summarized below. The previous SAXS measurement was using solid sample of Fmoc-F-F(His) nanofilaments. We have adjusted the horizontal and vertical coordinates of the SAXS data as shown in **Figure for review 1**.

In the last response, we corrected analysis based on the Guinier approximation and obtained the $R_g = 64.2 \pm 2.4$ nm. We agree with the reviewer that R and L cannot be determined from R_g . For a monodisperse scattering system of cylinders with length, L, and radius, R, the relationship among L, R and the radius of gyration, R_g , is given by [1]

$$R_g^2 = (R^2/2) + (L^2/12) \quad (1)$$

However, there is no simple correlation among these three structural parameters for the present polydisperse distribution system [2]. We fully agree with the reviewer's questioning. Meanwhile, such R_g value corresponds to the cylinder scatters with the dimensions of diameter $2R \sim 15$ nm and length $L \sim 220$ nm (based the Formula (1)) (It should be noted that R value was pre-assumed based on Cryo-EM and L value was calculated). Furthermore, we performed the size distribution analysis by fitting the SAXS data over the selected region using the total non-negative least square method [3] implemented in the Irena package [4] (**Figure for review 2**). Note that in this fitting analysis, for simplicity we made assumptions

that the minimum fibre diameter is 10 nm and the length is fixed at 220 nm based on our Cryo-EM observations and the above R_g analysis.

Figure for review 1 Guinier plot showing the analysis result of R_g using solid sample.

Figure for review 2 SAXS data fitting and the obtained fibre diameter distribution using solid sample (shown as the histogram plot).

But we did not show the computer fitted data we have done, which caused the confusion to the reviewer. In addition, we also noticed from Cryo-EM that the length of the cylinders

exceeds 500 nm. Such large length requires measurements down to very small q for proper Guinier analysis. Unfortunately the available SAXS setup has limited q range and could not satisfy such requirement. The rather small R_g (in comparison with the results in Cryo-EM) from previous Guinier analysis was also caused by the instrumental limitation. Thus, cross-sectional Guinier plots or Guinier plots may not be perfectly implemented based on current situation.

After carefully discussion with the expert in SAXS field, we realized that liquid sample is often used for nanofilaments measurement in SAXS, because nanofilaments maintain good dispersibility under aqueous condition. We therefore re-conducted the SAXS analysis with liquid samples as described in the detailed experimental methods and analyzed radius R using SAXS software. New data analysis has been carried out with model fitting in the whole q range instead of Guinier analysis on points with q smaller than 0.1 nm^{-1} .

In order to determine the fibre length L and its radius R , we performed a SAXS modeling analysis. The obtained SAXS intensities, $I(q)$, were fitted to the following equation using the SasView software package. The output of the 1D scattering intensity function for randomly oriented cylinders is thus given by:

$$I(q) = \frac{\text{scale}}{V} \int_0^{\pi/2} F(q, \alpha)^2 \sin \alpha \, d\alpha + \text{background} \quad (2)$$

Where

$$F(q, \alpha) = 2(\Delta\rho)V \frac{\sin(\frac{1}{2}qL\cos\alpha)}{\frac{1}{2}qL\cos\alpha} \frac{J_1(qR\sin\alpha)}{qR\sin\alpha} \quad (3)$$

α is the angle between the cylinder axis and the scattering vector q , and J_1 is the first order Bessel function. $(\Delta\rho)^2$ is the scattering contrast factor of the sample, $V=\pi R^2L$ is the volume of the fibre with a radius of R . L is the length of the cylinder, R is the radius of the cylinder.

As shown in **Figure for review 3**, we obtained a good SAXS fitting with the result of average $R = 8.18 \text{ nm}$, which is in agreement with the values observed by Cryo-EM (It should be noted that we found that regardless of how the length of L is changed, the fitted R remains almost unchanged, indicating the availability of R fitting).

Figure for review 3 SAXS data fitting using the cylinder model by SasView.

To ensure the rationality of our fitting, we utilized different software and considered the dispersity of the sample. The obtained SAXS intensities, $I(q)$, were fitted to the following equation using the Irena software package [4]:

$$I(q) = (\Delta\rho)^2 \Phi \int_0^\infty V_p(R) f(R) F(q, R)^2 dR \quad (4)$$

where $(\Delta\rho)^2$ is the scattering contrast factor of the sample, Φ is the scale factor, $V_p(R)$ is the volume of the fibre with a radius of R , $f(R)$ is the distribution function for the fibre radius R (approximated by a Schulz-Zimm size distribution here), and $F(q, R)$ is the cylinder form factor used for the fibre structure as observed with Cryo-EM, which is given by:

$$F(q, R) = 2(\Delta\rho)V_p \frac{\sin(\frac{1}{2}qL\cos\alpha) J_1(qR\sin\alpha)}{\frac{1}{2}qL\cos\alpha \quad qR\sin\alpha} \quad (5)$$

where α is the angle between the cylinder axis and the scattering vector q , and J_1 is the first order Bessel function. Through a least-square fitting, the pre-defined L of 150 nm estimated based on the Cryo-EM observation can be fitted, and the distribution of radius R (as shown in **Figure for review 4a**) and its average value are determined. As shown in **Figure for review 4b**, we obtained a good SAXS fitting with the result of average $R = 7.4$ nm, in agreement with the values given by Cryo-EM.

Figure for review 4 a. The fibre radius distribution determined from the SAXS model fitting.

b. An example fit of the SAXS data from the Fmoc-F-F(His) nanofilaments. The average radius of $R = 7.4$ nm is determined from SAXS fitting.

In the revised manuscript and supplementary information, we decided to show the SAXS experiments using liquid samples and the data for radius R value analysed by SaSView software (Marked in red). We are beginners in the field of SAXS characterizations and hope to make in-depth study on our peptide assembly using SAXS. We highly appreciate any questions and suggestions from the reviewers and hope to receive further guidance to improve our study.

Experimental with liquid sample

Small-angle X-ray scattering (SAXS) measurements

Small-angle X-ray scattering (SAXS) measurements were performed on the liquid samples of Fmoc-F-F(His) at a concentration of 3.75 mM loaded in a low noise flow cell using a Xenocs Xeuss 2.0 instrument equipped with a Dectris Pilatus 300k detector (pixel size $172 \mu\text{m}$) and a Cu-K α radiation source (wavelength $\lambda = 1.54189 \text{ \AA}$). 2D SAXS data were collected at an exposure time of 1200 s. The SAXS data of Silver Behenate standard were also collected and used to calibrate the sample-to-detector distance (2480 mm). The 2D SAXS data of the samples were reduced to obtain 1D scattering intensity profile $I(q)$, where q is the scattering vector ($q = 4\pi\sin\theta/\lambda$, 2θ is the scattering angle). The raw data were corrected for the sample

background (flow cell containing water) in the data reduction process using the XSACT software package.

References

- [1] T. Imae, T. Kanaya, M. Furusaka, N. Torikai, Neutrons in soft matter, John Wiley & Sons 2011.
- [2] A. Deschamps, F. De Geuser, On the validity of simple precipitate size measurements by small-angle scattering in metallic system, *Journal of Applied Crystallography* 44(2) (2011) 343-352.
- [3]. Michael Merrit and Yin Zhang, Technical report TR04-08, Department of Computational and Applied Mathematics, Rice University, Houston, Texas, 77005, USA.
- [4]. J. Ilavsky, P.R. Jemian, *Journal of Applied Crystallography*, 42 (2009) 347-353.

REVIEWERS' COMMENTS

Reviewer #3 (Remarks to the Author):

Authors removed the SAXS data taken on solid samples and added the corresponding data taken from solution samples. Although the analysis of the "Figure for review" 1 and 2 is wrong, this is not included in the manuscript and thus there is no problem. In the revised manuscript, they only put one figure describing the fitting of SAXS curve using a cylindrical model as FigS. 17. The fitting quality is nice at low-Q. The resultant radius of the fiber is consistent with that obtained by Cryo-EM, enhancing their results.

About the use of SasView, its reference should be made.

Reviewer #3 (Remarks to the Author):

Authors removed the SAXS data taken on solid samples and added the corresponding data taken from solution samples. Although the analysis of the "Figure for review" 1 and 2 is wrong, this is not included in the manuscript and thus there is no problem. In the revised manuscript, they only put one figure describing the fitting of SAXS curve using a cylindrical model as FigS. 17. The fitting quality is nice at low- Q . The resultant radius of the fiber is consistent with that obtained by Cryo-EM, enhancing their results.

About the use of SasView, its reference should be made.

Response: Thanks for the positive comments and suggestions of the reviewer. We have added two references (reference 1, 2) in the manuscript (corresponding to reference 37, 38) for providing the use of SasView. Meanwhile, we have added the website (<https://www.sasview.org/>) where the instructions of SasView can be found.

References

1. McDowall, D., Adams, D. J., Seddon, A. M. Using small angle scattering to understand low molecular weight gels. *Soft Matter*. **8**, 1577-1590 (2022).
2. Doucet, Mathieu, Cho, Jae, H., Alina, Gervaise, Attala, Ziggy, Bakker, Jurrian, Bouwman, Wim, Butler, Paul, Campbell, Kieran, Cooper-Benun, Torin, Durniak, Celine, Forster, Laura, Gonzales, Miguel, Heenan, Richard, Jackson, Andrew, King, Stephen, Kienzle, Paul, Krzywon, Jeff, Nielsen, Torben, O'Driscoll, Lewis, et al. (2020). SasView version 5.0.3. Zenodo.